# Tuning Li occupancy and local structures for advanced Co-free Ni-rich positive electrodes

Hang Li[1,2] ✉, Hao Liu[2], Shunrui Luo[3], Jordi Arbiol [3,4], Emmanuelle Suard [5], Thomas Bergfeldt [2], Alexander Missyul [6], Volodymyr Baran[7], Stefan Mangold[8], Yongchao Zhang[3], Weibo Hua [9], Michael Knapp [2], Helmut Ehrenberg [2], Feng Pan [1] ✉ & Sylvio Indris [2,10] ✉

Structure evolution and surface reactivity have long been regarded as the most crucial points for studying Ni-rich positive electrodes for Li-ion batteries. Unfortunately, the influence of Li occupancy as a single factor on electro-chemomechanical stability has been overlooked and is missing, owing to the challenge of Li determination in the lattice. Here, a comprehensive analysis reveals different Li occupancies and related structural domains (Ni/Li exchange, $Li_aXO_b$, Li/Mn/X(Ni) ordering domains, $X = Nb^{5+}$, $W^{6+}$, and $Mo^{6+}$) by using a combination of Li-sensitive characterization techniques. By introducing a Li-regulation strategy, the relative ratio of each domain is effectively tuned in the Ni-rich positive electrodes. Through tuning, two specific positive electrodes are designed, exhibiting notable improvement in battery cyclability. The specific Li structural units induce significant changes in redox mechanisms. This Li-occupancy-tuning approach highlights the necessity of focusing on Li distribution and opens up ideas for designing advanced Ni-rich positive electrodes with high durability.

Bulk and surface instabilities present in electrode materials usually induce performance fading of the batteries[1–4]. Modification strategies, such as doping and surface coating, are proposed for mitigating such instabilities and improving the electrochemical capability of these materials[5–9]. Unfortunately, the stabilization for Ni-rich layered oxides is challenging since increasing the Ni content pushes the positive electrode chemistry to their parent material, $LiNiO_2$. It deeply suffers from critical surface reactivity of $Ni^{3+/4+}$ and bulk crystal structure degradation, thereby remaining unsuitable for commercial application[10–12].

The Li occupancy inside the positive electrode lattice tends to notably influence the crystal structure of Ni-rich materials and their electrochemical stability[13–15]. Unfortunately, compared with apparent structure and phase changes during modifications, the corresponding scenario of Li occupancy falls short of being focused on, owing to the intrinsic weak sensitivity for Li when using X-ray-based characterization techniques. Ideally, Li and transition metals (TM) are placed separately on 3b and 3a sites, respectively, in the unit cell, forming alternating Li and TM layers[16]. Unfortunately, structural defects are impossible to avoid in layered oxides of Ni-rich positive electrodes. Regarding Li, such defects include Li/Ni exchange (or Li/Ni disordering) and extra Li located in TM layers[17,18]. The former one mainly originates from the introduction of foreign high-valence dopants (like $Mn^{4+}$) into the lattice, to realize the local charge compensation[19].

[1]School of Advanced Materials, Peking University Shenzhen Graduate School, Shenzhen, China. [2]Institute for Applied Materials (IAM), Karlsruhe Institute of Technology (KIT), Eggenstein-Leopoldshafen, Germany. [3]Catalan Institute of Nanoscience and Nanotechnology (ICN2), CSIC and BIST, Campus UAB, Bellaterra, Barcelona, Catalonia, Spain. [4]ICREA, Barcelona, Catalonia, Spain. [5]Institute Laue-Langevin (ILL), Grenoble, France. [6]CELLS-ALBA Synchrotron, Barcelona, Spain. [7]Deutsches Elektronen-Synchrotron (DESY), Hamburg, Germany. [8]Institute for Photon Science and Synchrotron Radiation, Karlsruhe Institute of Technology (KIT), Eggenstein-Leopoldshafen, Germany. [9]School of Chemical Engineering and Technology, Xi'an Jiaotong University, Xi'an, Shanxi, China. [10]Applied Chemistry and Engineering Research Centre of Excellence (ACER CoE), Université Mohammed VI Polytechnique (UM6P), Ben Guerir, Morocco. ✉e-mail: 2206393335@pku.edu.cn; panfeng@pkusz.edu.cn; sylvio.indris@kit.edu

Recently, a Li occupation structure, namely Li/Ni superlattice units, has been proposed, which could be regarded as an ordered Li/Ni exchange[8,20,21]. The latter one is typically induced by the extra usage of Li during synthesis, sometimes forming $Li_2MnO_3$ domains (if Mn is used as TM) where each Li at TM sites is surrounded by six $Mn^{4+}$ dopants (honeycomb structure)[22]. A recent study shows that a Li(Ni)/W ordering (inverse honeycomb) in the TM layers can be formed when $W^{6+}$ is introduced into the $LiNiO_2$ lattice[23]. Similar to the $Li_2MnO_3$ domains, that special structural unit activates an additional oxygen redox mechanism. Note that each above structural unit plays an important role in the structural stability and electrochemical performance of the positive electrode materials. For each unit, identification and further optimization of the content is a potential way to obtain high-performance positive electrode materials. Therefore, the potential ways to tune performance are multiple, depending on which specific structural units are optimized.

To verify the above ideas, in this article, we propose that the Li occupancy in each domain can be tuned by adjusting the Li incorporation during synthesis. Herein, $Mn^{4+}$ and multiple high-valence-state dopants ($Nb^{5+}$, $W^{6+}$, and $Mo^{6+}$) are introduced into the Ni-rich layered oxide lattice simultaneously to create environments with various Li occupancy (such as Li/Ni exchange, and Li/Mn/X(Ni) local structures). The introduction of multiple dopants also helps increase the overall doping content and introduce entropy stabilization for better cycling stability[24]. Rational control over Li content is performed when synthesizing such multi-doped positive electrodes from its hydroxide precursors. The tuning of Li occupancy is based on two principles: high-valence elements tend to combine with Li and segregate from the lattice, forming Li-rich oxides ($Li_aXO_b$, X = $Nb^{5+}$, $W^{6+}$, and $Mo^{6+}$) separately[13,25,26];. At the same time, extra Li helps suppress the Li/Ni exchange and leaves Li in TM layers, forming certain Li/Mn/X(Ni) ordering domains. With the nominal Li content ($x$) increasing from 1.02 to 1.20, Li/Ni exchange is suppressed, accompanied by the increased content of $Li_aXO_b$ and the Li/Mn/X(Ni) ordering domains.

Based on such knowledge, our attention was drawn to the two positive electrodes with nominal Li contents of 1.08 and 1.20, which are distinguished by a reasonable content of Li/Ni exchange (1.08) and $Li_aXO_b$ and the Li/Mn/X(Ni) ordering domains (1.20), respectively. Both materials demonstrate significantly improved durability during battery cycling, but for different reasons: the former contributes to the effect of robust Li/Ni exchange stabilization. The latter can be explained by the introduction of the additional oxygen redox, which alleviates the lattice change and increases the stability of the positive electrodes at higher charging potentials. These findings highlight multiple options for optimizing the performance of a single Ni-rich positive electrode through tuning Li occupation and exhibit practical ideas for developing advanced high-performance Ni-rich positive electrodes.

## Results
### Li occupancies
$Li_xNi_{0.83}Mn_{0.13}Mo_{0.02}Nb_{0.01}W_{0.01}O_2$ (HD-LNMO-102,108,114, and 120, where HD denotes high-valence doped, the numbers represent nominal Li content 1.02, 1.08, 1.14, and 1.20) was developed by using a typical co-precipitation method (see the Experimental section). To investigate the influence of Mo, Nb, and W dopants on structure, the reference material $LiNi_{0.83}Mn_{0.13}Co_{0.04}O_2$ (RM) was synthesized, and conventional Co replaced those high-valence-state dopants. The element compositions were analyzed using inductively coupled plasma-optical emission spectroscopy (ICP-OES), displaying an overall agreement with the designed stoichiometry (Table S1). Synchrotron X-ray diffraction (SXRD) and neutron diffraction (ND) patterns were collected from the samples to probe the structural evolution as a function of Li content (Fig. 1a, b). All the materials demonstrate an identical layered oxide structure with *R-3m* symmetry. No secondary phase is obtained, except for the tiny reflections from vanadium containers in ND patterns. As shown in enlarged views, the peak positions of the 110 and 214 reflections shift towards higher 2θ values with increasing Li content, suggesting a gradual decrease in lattice parameters. There is apparent peak splitting for HD-LNMO-120, an indicator of cation ordering and less site disorder.

To determine the site occupancies of the Li (3b) and TM (3a) sites, combined Rietveld refinement of structural models was performed against both SXRD and ND data (Supplementary Note 1, Table S2–S5). The combination of both data sets is especially suitable for Ni-rich positive electrode materials: compared to X-rays, neutron scattering has significantly improved sensitivity for Li, therefore, in the structural model two occupancy parameters, $Ni_{Li}$ (additional Ni on the Li sites) and $Li_{TM}$ (additional Li on the TM sites) were allowed to be refined in parallel[27]; The total contents of Ni and Li are constrained by their values obtained from ICP-OES. The high intensity and negligible peak profile contributions of the instrument in SXRD patterns allow for the precise determination of structural parameters. An example of the combined refinement is provided in Figs. 1c and 1d for SXRD and ND patterns of HD-LNMO-120. Fig. S1 displays the refinements of other samples. The resulting site occupancies and their evolution with varying Li content ($x$) are displayed in Fig. 1e. With increasing $x$, $Li_{TM}$ generally increases, while $Ni_{Li}$ keeps decreasing. Notice that antisite defects exist after Mn doping, where neighboring Li and Ni exchange their occupied sites. Therefore, the amount of $Li_{TM}$ matched by an equal amount of $Ni_{Li}$ represents Li/Ni antisite defects in HD-LNMO. As $x$ increases, the decrease of $Ni_{Li}$ indicates a gradual suppression of Li/Ni exchange, in agreement with the peak splitting for HD-LNMO-120. Since the Li/Ni exchange is mostly based on $Ni^{2+}$, less exchange is also reflected by a gradual increase of the Ni oxidation state on increasing $x$. The changes in the average valence state of Ni are confirmed by the X-ray absorption spectroscopy (XAS) experiment (Fig. S2 and Supplementary Note 2). Furthermore, a decreasing amount of $Ni^{2+}$ is also suggested by the drop of the lattice parameters with HD-LNMO-102 to 120 (Fig. S3). This can be rationalized with the decreasing radii of the Ni-ions when oxidized from +2 to +3 state (r($Ni^{2+}$) = 0.69 Å, r($Ni^{3+}$) = 0.56 Å)[28]. An increase in the $c/a$ ratio (Fig. S3) also hints at a gradually lower Li/Ni exchange[29]. It should be noted that at this stage Li/Ni exchange includes both Li/Ni disordering and ordering (Li/Ni superlattice) because they cannot be distinguished by the model applied in the refinement.

Sole Li is defined as $Li_{sole} = Li_{TM} - Ni_{Li}$, which represents excess Li at TM layers not induced by the Li/Ni exchange effect. As visualized in Fig. 1e, $Li_{sole}$ continuously increases upon increasing $x$, corresponding to the Li excess in the elemental analysis ($Li_{sole}$ = Li-1 in Table S1). The exact local structure of this Li-rich domain will be discussed in the following part. The comparison in structure between RM and HD-LNMO-108 (the same nominal Li content) shows that the introduction of Mo, Nb, and W induces a higher degree of Li/Ni mixing since additional $Ni^{2+}$ ions are formed (Fig. S4, S5, and Table S6). The decreased average oxidation state of Ni in RM (compared to HD-LNMO-108) can also be checked by the corresponding X-ray absorption near edge structure (XANES) data. The doping of Mo, Nb, and W increases the lattice parameters and decreases the hexagonality ($c/a$ ratio).

$^6Li$ solid-state nuclear magnetic resonance (ss-NMR) was performed on $^6Li$-enriched samples to comprehensively probe the different Li local environments and track their content changes among samples (Fig. 2a). Detailed discussion on $^6Li$-enriched samples and the comparison with normal $^7Li$ samples are provided in Supplementary Note 3 and Fig. S6. A detailed deconvolution was conducted on the $^6Li$ spectra in Fig. 2b and Table S7. There are five resonance peaks in total, which are centered at around 605, 296, 260, 4, and -21 ppm, respectively (labeled as A–E in Fig. 2b). For Ni-rich materials, the $^6Li$ NMR shifts are dominated by the Fermi contact interaction, which originates from the delocalization of unpaired electron spin density from TM $d$ orbitals to the Li nucleus via the bridging oxygen atoms[30].

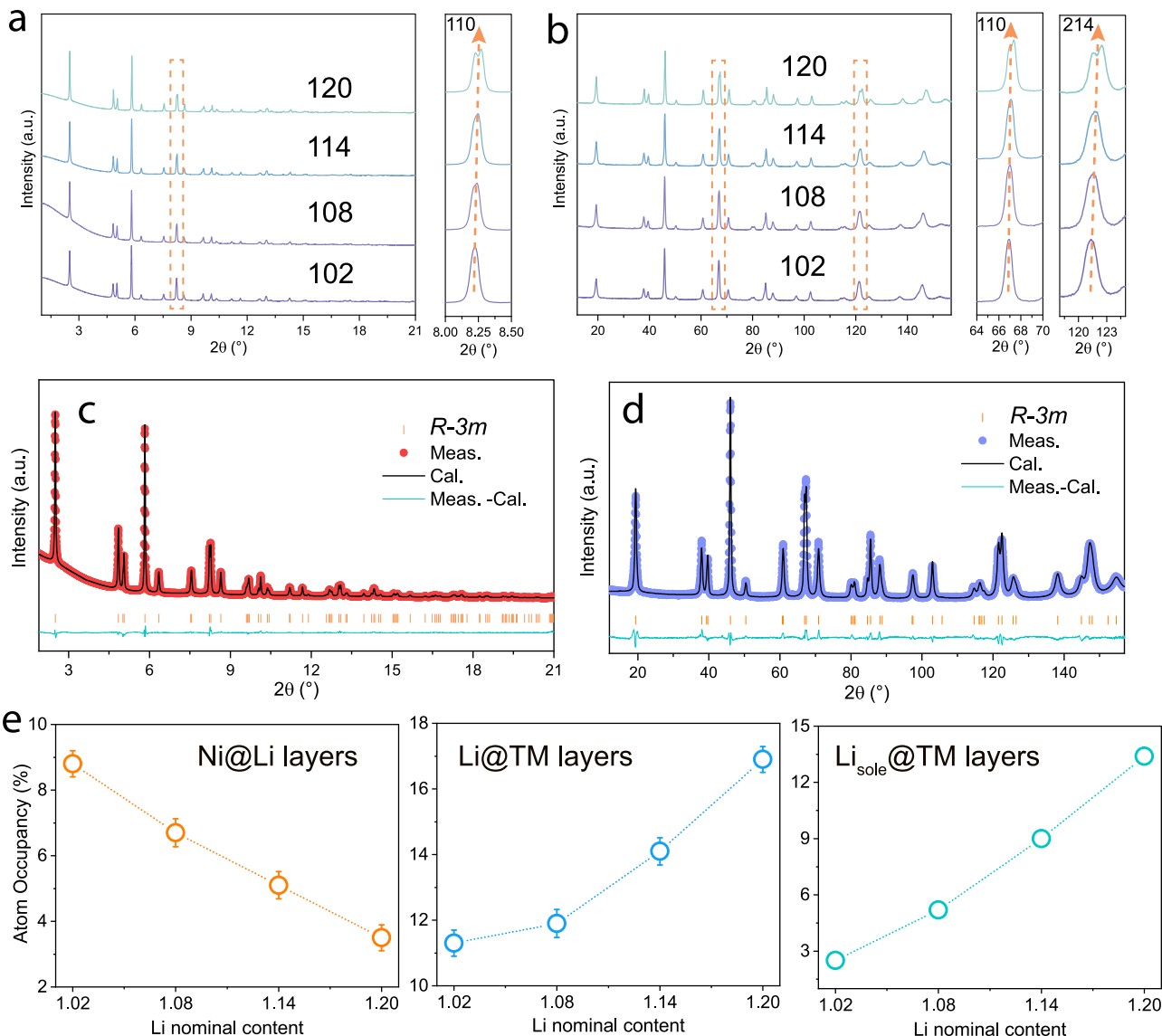

**Fig. 1 | Bulk structural characterization of HD-LNMO-102, 108, 114, and 120.** Stacked plots of (**a**) SXRD patterns and (**b**) ND patterns collected from the as-synthesized materials. The wavelengths are 0.2073 Å and 1.594 Å for SXRD and ND data, respectively. Example of simultaneous Rietveld refinements of HD-LNMO-120 to (**c**) SXRD pattern and (**d**) ND pattern. **e** Refined atomic parameters as a function of Li content, including Ni in the Li layers ($Ni_{Li}$), Li in the TM layers ($Li_{TM}$), and $Li_{sole}$ in the TM layers (obtained by subtracting $Ni_{Li}$ from $Li_{TM}$). The error bars in (**e**) are produced during the combined refinement.

The contribution from each adjacent paramagnetic TM ion (here, Ni and Mn) is additive to the overall $^6$Li NMR shifts and the magnitude is highly correlated to the TM oxidation state ($Ni^{2+}$, $Ni^{3+}$) and the angle of the TM-O-Li bond[31–33]. Diamagnetic ions, including $Ni^{4+}$, $Mo^{6+}$, $Nb^{5+}$, and $W^{6+}$, are believed to only induce minor shifts.

Regarding the spectra, peaks A and C collectively describe the broad signal starting from around 1200 to −250 ppm, which mainly corresponds to the $Li^+$ in the Li layer ($Li_{Li}$). The broadening can be rationalized by mainly two aspects: the partial reduction of Ni from 3+ to 2+ (to maintain charge balance for the $Mn^{4+}$)[32,33]; a mixture of diverse Li local environments, such as Li surrounded by pure Ni or a combination of Ni and Mn[34]. The low signal at 296 ppm (peak B) in all spectra is most likely due to a small amount of $Li_{TM}$ surrounded by one $Mn^{4+}$ in the TM layers. Since $Li_{TM}$ has almost only Li neighbors in the layers above and below (Li layer), therefore the chemical shift of $Li_{TM}$ is dominated by the 6 TM-O-Li 90° neighbors in the TM layers. Yoon reported that for Li-rich Mn-rich materials Li in the TM layers surrounded by six $Mn^{4+}$ (honeycomb structure) has an NMR shift of 1560 ppm[22]. Owing to the Ni-rich Mn-poor nature of our materials, Li in the

TM layers may have only one $Mn^{4+}$ neighbor, corresponding to one-sixth of the NMR shift (290 ppm). For the other five neighbors, the option could be a mixture of Ni, Nb, Mo, and W (i.e. Li/Mn/X(Ni) ordering units). Those ions only induce minor NMR shifts[26]. Note that $Ni_{Li}$ defects are predicted to also contribute to the positive shift if they are placed close to the above-mentioned $Li_{TM}$ sites[33]. When the Li content $x$ increases in HD-LNMO from 102 to 120, the proportion of Li belonging to this $Li_{TM}$ gradually increases, accounting for 0.3%, 1.4%, 2.0%, and 4.3% of the total Li signal in the spectra (see Fig. 2c). As mentioned earlier, $Li_{TM}$ includes both Li/Ni exchange defects and $Li_{sole}$ at TM layers, and only $Li_{sole}$ exhibits the same increasing trend as that of the NMR signal. Therefore, peak B belongs to the sole Li in the combined refinement result. A $Li_{sole}$ near the $Mn^{4+}$ neighbors helps maintain charge balance. The sharp diamagnetic signal (peak D) at around 4 ppm corresponds to the Li in the $Li_aXO_b$ (X = $Mo^{6+}$, $Nb^{5+}$, and $W^{6+}$) phase[26]. The presence of a diamagnetic contribution in NMR spectra of Ni-rich materials usually originates from impurities, for example, LiOH and $Li_2CO_3$, owing to the Li excess during synthesis[31]. However, their reflections are not found in the corresponding SXRD or

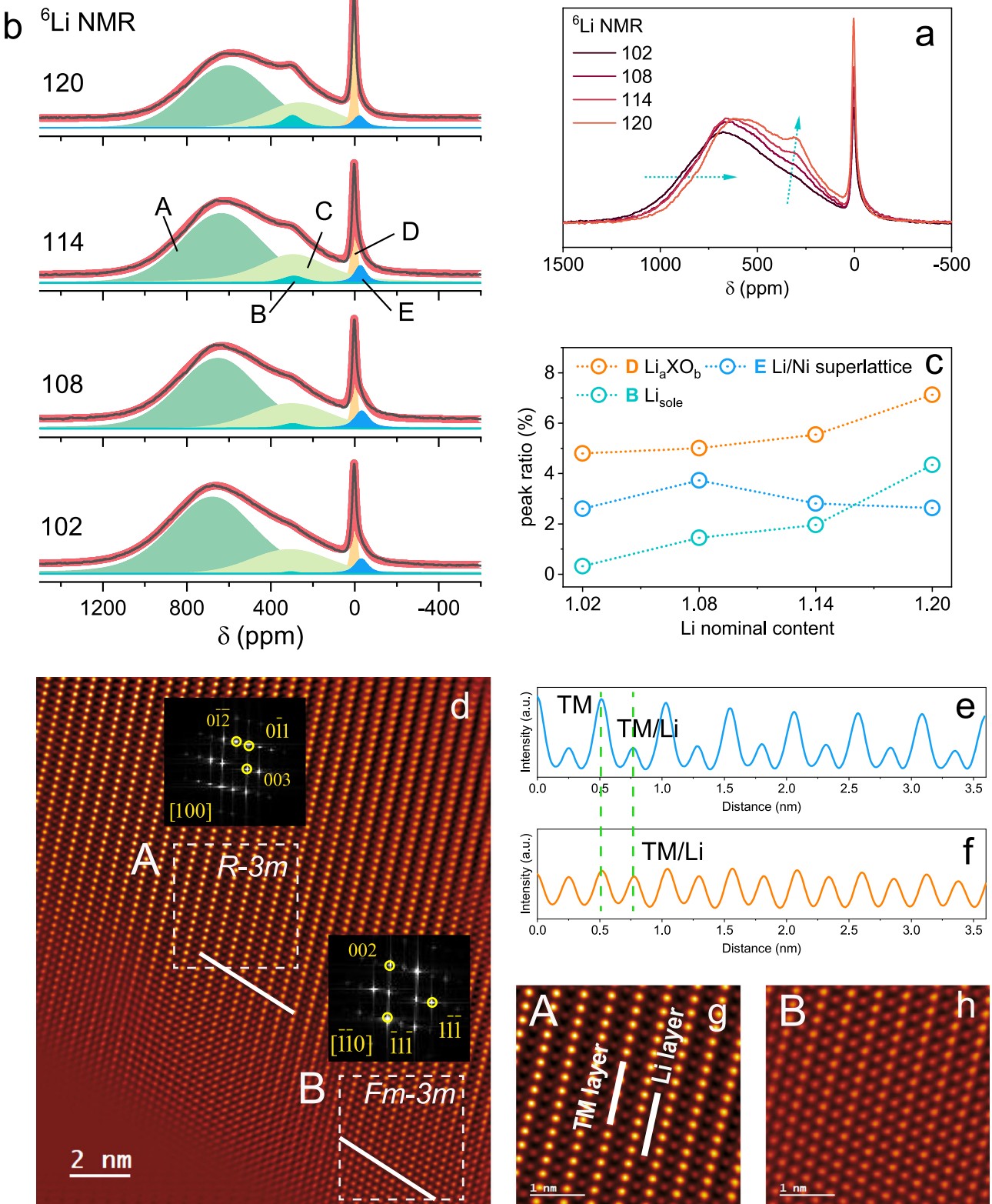

**Fig. 2 | Local structural characterization of HD-LNMO-102, 108, 114, and 120.**
**a** ⁶.Li MAS-NMR spectra. **b** fitting of the NMR spectra. Different peaks are labeled with A, B, C, D, and E. **c** peak ratio (%) of different species ($Li_aXO_b$, Li/Ni superlattice, and $Li_{sole}$). **d** HAADF STEM image showing the atomic structures of the HD-LNMO-108. The inset is the FFT patterns from the selected regions of A and B,

corresponding to layered (*R-3m*) and rock salt (*Fm-3m*) structures, respectively. **e**, **f** Line intensity profiles from lines 1 and line 2 in (**d**). TM or Li are labeled based on the difference in the relative intensity. **g**, **h** Enlarged HAADF-STEM images from the marked regions A and B in (**d**). TM and Li layers are indicated in (**g**).

ND patterns. High-valence elements (larger than 4+) are reported to tend to crystallize as the Li-enriched cation-ordered rock salt structures (such as $Li_3NbO_4$ and $Li_4MoO_5$). $Ni^{2+}$ can substitute for Li and X, forming a rock salt-type structure with space group $Fm-3m$[35,36]. As displayed in Fig. 2c, the ratio of the signal D increases with increasing Li content (4.8%, 5.0%, 5.6%, and 7.1%), suggesting that the formation of the $Li_aXO_b$ phase is promoted by the increasing amount of Li.

Up to now, the Li signals from the Li/Ni exchange are still missing. Herein, a sub-classification into Li/Ni disordering and Li/Ni ordering (superlattice) is necessary for further discussion. For Li/Ni disordering, it is related to the $Mn^{4+}$ in the TM layers ($Ni^{2+}$ ions are created to keep electrical neutrality), so these exchanged Li ions are believed to have $Mn^{4+}$ neighbors in a 90° $Mn^{4+}$-O-Li configuration[37]. On the contrary, the Li/Ni superlattice structure is induced by the diamagnetic high-valence state elements[21]. To maintain charge neutrality, $Mn^{4+}$ should avoid these local domains. Therefore, the signal of Li in Li/Ni disordering is expected to appear in the positive ppm region, while the signal from Li in superlattice domains is believed to appear in the negative ppm region because Li is only in 90° $Ni^{2+/3+}$-O-Li configurations. Peak A and C are predicted to contain the Li signals from Li/Ni disordering, and peak E is probably due to the Li in the Li/Ni superlattice domains. As plotted in Fig. 2c, HD-LNMO-108 has the largest relative intensity of signal E among all samples, indicating the highest fraction of the superlattice domains in this material. Looking back to the NMR spectra in Fig. 2a, fewer signals are located on the left-hand side (700–1200 ppm) with increasing Li content $x$. This signal decrease corresponds to less $Ni^{2+}$ in the lattice since 180° $Ni^{2+}$-O-Li induces larger chemical shifts than 180° $Ni^{3+}$-O-Li, in agreement with the XANES results and the evolution of Ni/Li exchange[32,33]. On the contrary, the intensity on the right-hand side (0–700 ppm) region keeps increasing for samples from 102 to 120, reflecting that an increasing amount of sites are occupied by Li rather than $Ni^{2+}$ in the Li layers, consistent with the decreasing trends of $Ni^{2+}$ and Li/Ni exchange. The $^6Li$ NMR spectrum of the reference material (RM) was collected and compared with the one of HD-LNMO-1.08 (Fig. S7). Owing to the reduced Li/Ni exchange in RM, a stronger signal intensity was obtained in the region of 240–900 ppm in comparison with the counterpart. Interestingly, such strong intensity obscures the resonance at 296 ppm from $Li_{sole}$ with $Mn^{4+}$ neighbors for RM, although a strong signal for such structure is expected in the spectrum because of the larger $Li_{sole}$ value (8.5%) obtained from the refinement. Furthermore, as suggested by diffraction patterns, the sharp signal at 3 ppm is assigned to $Li_2CO_3$ in the RM material (not to $Li_aXO_b$ or superlattice), hence the apparent difference in the peak shape and intensity between the spectra of RM and HD-LNMO-108 can be well understood.

To further determine the atomic structures of the HD-LNMO materials, a high-angle annular dark field scanning transmission electron microscope (HAADF-STEM) image of HD-LNMO-108 was acquired (Fig. 2d). Two types of local structures are identified by their corresponding fast Fourier transformation (FFT) patterns from the selected areas labeled with A and B, indexed to layered ($R-3m$) and rock-salt ($Fm-3m$) domains, respectively[6]. The two domains exhibit different characteristic periodic atomic arrangements in the TM and Li layers, as confirmed by the height profiles of lines 1 and 2 (Fig. 2e, f). The marked regions A and B in Fig. 2d are enlarged and shown in Fig. 2g, h. The Li layer is typically not visible in the HAADF-STEM image because of the low atomic scattering factor of Li[21]. Interestingly, successive patterns of dots (tiny green balls in Fig. 2g) in Li layers are observed, although their brightness is lower than that of the dot patterns in TM layers, as also confirmed by the height profiles in Fig. 2e. The enhanced contrast in the Li layers implies that a few Li atomic sites are occupied by Ni through Li/Ni exchange[38]. It is reported that a periodic contrast in Li and TM layers in Li/Ni superlattice domains (Ni alternately occupies the Li layer and vice versa for Li in the TM layer) can be detected owing to the ordered Li/Ni exchange after high-valence dopant doping[20].

However, such a special contrast ordering is not observed in our HAADF-STEM image because the samples were not thinned by a focused ion beam (FIB) as described in the literature[6,20,21]. Regarding the HAADF-STEM image with a large sample thickness, the atomic contrast reflects the contrast accumulation from atoms in all layers perpendicular to the beam. Therefore, a superlattice domain probably exists, but the contrast contribution from multiple atoms covers the characteristic periodic contrast from Li/Ni exchange orderings. In the near-surface region, the contrast in the TM layer gradually decreases, confirmed by the intensity difference in the intensity profiles (Fig. 2e, f). Such a change implies that Ni and Mn in the TM layer are partly replaced with Li, forming Li-enriched $Li_aXO_b$ rock-salt ($Fm-3m$) domains, in line with the discussion in the NMR part. Depending on the specific contents of Li, X, and $Ni^{2+}$ in $Li_aXO_b$, such domains could also be indexed to $R-3m$ layered structure (see discussion on the HAADF images of HD-LNMO-120 (Fig. S8) in Supplementary Note 4). Since all the reflection peaks from the $Fm-3m$ phase will be overlapped by the stronger reflections from the bulk $R-3m$ phase at the same positions, only a single-phase model is used during the combined refinement for the sake of simplicity and repeatability. The $Li_{sole}$ given by the refinement could serve as a whole description for the $Li_aXO_b$ phase as well as the Li/Mn/X(Ni) ordering domains. No clear domain boundary is formed owing to the high structural similarity between the two phases.

## Occupation-tuning strategy

With the knowledge gained from combined refinements, NMR, and HAADF STEM, the whole Li occupation situation in HD-LNMO and the evolution of each structural unit during the Li tuning process is possible to analyze. Figure 3 shows the schematic images of four different Li-containing structures, i.e. Li/Ni disordering, Li/Ni ordering (superlattice), Li/Mn/X(Ni) ordering, and $Li_aXO_b$. Li in the Li layers, as the primary occupation situation for Li, is not displayed here. The changes in the relative content for each structure are also summarized just below the corresponding schematic drawings. The plot is based on the input from the above analysis. Note that the content of Li/Ni disordering is calculated by subtracting the value of the superlattice (NMR) from the value of the Li/Ni exchange (combined refinement). To understand such evolutions, it is essential to figure out the existing manners to realize the charge balance in the materials. As $Mn^{4+}$ is more positively charged than $Ni^{3+}$ in TM layers, there are mainly two ways to balance the charge: the reduction of $Ni^{3+}$ to $Ni^{2+}$ and the incorporation of extra Li in the TM layers. They correspond to the Li/Ni disordering and $Li_{sole}$ with $Mn^{4+}$ neighbors in the previous discussion. Note that $Li_{sole}$ with $Mn^{4+}$ is a more efficient method for charge compensation since 1 Li/Ni disordering balances 1 $Mn^{4+}$ and 1 $Li_{sole}$ with $Mn^{4+}$ balances 2 $Mn^{4+}$. Regarding the high-valence elements, taking $Mo^{6+}$ as an example, the presence of 1 $Mo^{6+}$ in the TM layers induces 3 charge-compensating $Li/Ni^{2+}$ exchanges in the lattice. A large number of exchanges are believed to form a so-called superlattice structure. Similar to $Mn^{4+}$, the parallel way to decrease the charge is to place extra Li in the TM layers close to $Mo^{6+}$.

Such a balancing manner gives rise to the Li/Mn/X(Ni) ordering and the phase formation of $Li_aXO_b$ and is also more efficient than Li/Ni exchange. From the perspective of charge neutrality, higher Li content can promote the formation of the Li/Mn/X(Ni) ordering and $Li_aXO_b$, hence suppressing the complementary charge-balancing manners, i.e., Li/Ni disordering and Li/Ni superlattice, respectively. With the Li content changing from 102 to 108, the relative content of Li/Ni disordering is decreasing while both the Li/Mn/X(Ni) ordering and $Li_aXO_b$ exhibit increasing tendencies. Interestingly, an increasing trend is obtained for the superlattice, contrary to the predicted overall tendency. Owing to the limited content of Li in 102, part of the $Ni^{2+}$ ions surrounding X may not exchange with Li but stay in the TM layers, not forming the Li/Ni superlattice. From 108 to 120, the trends in the relative content of all four different Li occupation sites meet our hypothesis, which is based

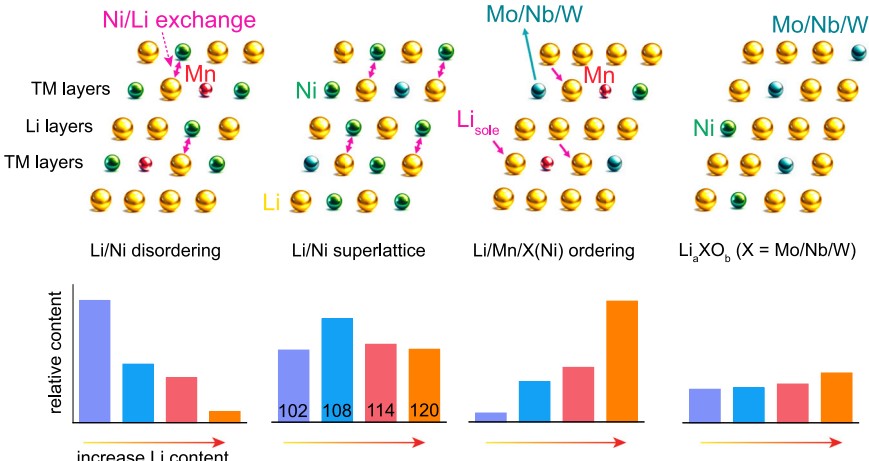

**Fig. 3 | Occupation tuning strategy.** Schematic illustration of the different structural models for HD-LNMO materials. From left to right: Li/Ni disordering, Li/Ni ordering (superlattice), $Li_{sole}$ at TM layers (with $Mn^{4+}$ neighbors), and $Li_aXO_b$ (X could be Mo/Nb/W). The gold, green, blue, and red balls indicate atoms of Li, Ni, X(Mo/Nb/W), and Mn, respectively. The double arrows represent locations where Li/Ni exchanges happen, and the single arrows refer to the $Li_{sole}$ at TM layers. When increasing the Li content, the evolutions in the relative content of each structural motif are indicated in the bar charts just below the corresponding models.

on the charge-compensation mechanism. With increasing Li content, Li/Mn/X(Ni) ordering and $Li_aXO_b$ gradually replace Li/Ni exchange, becoming the main method to achieve charge balance. By simply controlling the Li content during synthesis, positive electrodes with different Li occupations can be designed: A Ni-rich cathode featuring a high-content Li/Mn/X(Ni) ordering (more Li) or a Ni-rich cathode featuring a moderate degree of Li/Ni exchange (less Li).

**Electrochemical properties**
Electrochemical properties were characterized in half cells to demonstrate the influence of Li occupation on battery performance. The first charge-discharge profiles at 18 mA g⁻¹ are displayed in Fig. 4a. Compared with the reference material (RM), although the initial discharge capacity of 102 is slightly decreased from 200 to 196 mAh g⁻¹, probably owing to the larger extent of Li/Ni disordering[39], upon further increasing the Li contents, much higher specific capacities (204–206 mAh g⁻¹) than RM could be delivered. In addition, the initial Coulombic efficiency of all HD-LNMO positive electrodes outperforms that of RM, indicating enhanced (de)lithiation reversibility. As shown in the corresponding dQ/dV (Q is the capacity and V is the voltage) curves (Fig. 4b), the shape of these dQ/dV peaks does not evolve too much except for some details. A careful discussion on this point will be conducted later. Figure 4c, d show the long-term cycling performance of HD-LNMO in half-cells at room temperature (25 °C). The corresponding information including specific discharge capacity and Coulombic efficiency can be checked in Fig. S9. The cyclability of HD-LNMO strongly depends on the Li content x. For 102 and 114, they exhibit faster capacity fading than that of RM. In sharp contrast, HD-LNMO-108 and 120 show improved capacity retention at different cutoff voltages, for example, 93.0% and 91.4% capacity retentions after 100 cycles at 2.5–4.3 V (versus Li/Li⁺) and 91.8% and 92.2% capacity retentions after 80 cycles at 2.5–4.4 V (versus Li/Li⁺). Note that electrolyte depletion and heterogeneous solid electrolyte interface (SEI) may accelerate the fading process of the half cells. These values are similar to the corresponding ones from RM (90.4% and 89.0% for cutoff voltages of 4.3 V after 100 cycles and 4.4 V after 80 cycles, respectively). Note that RM is considered a reference material with good stability, owing to the doping of Co, less Ni/Li disordering, and potential stabilization effect from Li in the TM layers. Moreover, the cyclability of HD-LNMO-120 can be further improved when the lower cut-off voltage is changed from 2.5 to 2.0 V (see Fig. S10, increasing the

capacity retention from 91.4% to 93.5% after 100 cycles). Such an increase will be discussed in the next part.

**Structural stability**
Figure 5a compares the enlarged dQ/dV profiles of the first charge-discharge curves at ~~C/10~~ 18 mA g⁻¹ (full comparison in Fig. S11). As expected, the sharpness of the plateaus at around 4.25 V gradually increases with the increasing Li content, indicating a less mitigated H2-H3 phase transition, owing to the decreased Li/Ni disordering in the materials. Interestingly, the sample 120 does not conform to this trend. A detailed comparison between the curves of 108 and 120 is displayed in Fig. 5b. Compared with 108, the H2-H3 redox peak in 120 is obviously flattened, although a much smaller value for Li/Ni disordering is confirmed in the latter material (6.4% *vs* 3.2%). Moreover, a new peak appears at around 2.84 V for 120 during discharge, which is absent in the corresponding profile of 108. Such behaviors indicate a different redox mechanism for 120, which will be further discussed below.

To probe the accurate phase transition behaviors on Li removal/insertion, in situ SXRD was performed and the patterns in the 003 reflection region are displayed in Fig. 5c–e and Fig. S13. Overall, all positive electrodes exhibit a solid-solution behavior during the (de) lithiation process since no splitting appeared for the 003 reflections in the voltage range of 2.0–4.4 V[31]. Herein, the discussion is focused on structural evolutions in selected positive electrodes (HD-LNMO-108, 114, and 120) owing to their difference in cycling performance. The analysis of the other positive electrodes can be found in Figs. S11 and S12. At the end of charge, the 003 reflection of RM shifts by 0.09° to higher 2θ angles (Fig. S12) while the corresponding values are 0.03°, 0.02°, and −0.02° for HD-LNMO-108, 114, and 120, respectively, suggesting substantially suppressed phase transitions for HD-LNMO positive electrodes. Rietveld refinements against in situ diffraction patterns were processed to further compare the evolution of the lattice parameters during charging. Note that states of charge (SoCs) are applied here rather than charge cut-off voltage to reflect the actual amount of Li removal from the lattice. The maximum changes in the c-axis lattice parameter (Δc) and the lattice volume (ΔV) are 1.06% and 3.95% for 108 and 1.08% and 3.99% for 114. In clear contrast, at the same SoC (81%), the changes in (Δc) and (ΔV) are only 0.50% and 2.78% for 120. Supplementary Note 5 and Fig. S14 display the bond length and Bond Valence Calculation (BVS) information based on the Rietveld refinement results. A comparison between 120 and RM electrodes on

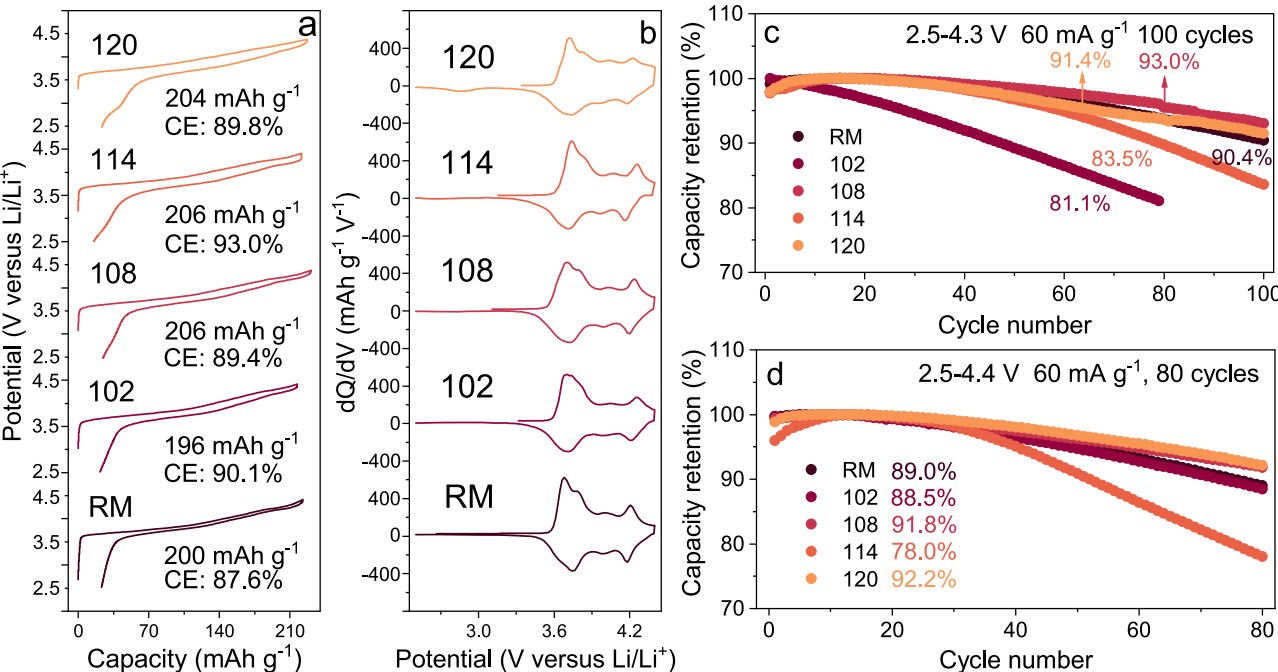

**Fig. 4 | Electrochemistry of HD-LNMO positive electrodes in half cells (at room temperature (25 °C) and with an active material mass loading of around 4 mg cm⁻²). a** First cycles of HD-LNMO with their first discharge capacities and first-cycle Coulomb efficiency at 18 mA g⁻¹ in the voltage range of 2.5–4.4 V. **b** Corresponding dQ/dV plots. **c** Capacity retentions of HD-LNMO cathodes cycled at 60 mA g⁻¹ in the voltage range of 2.5-4.3 V. **d** Capacity retentions under a higher charging cut-off voltage of 4.4 V between 1 and 80 cycles.

lattice changes during discharge is conducted to investigate the potential different redox mechanism for 120 (Fig. S15). RM exhibits a constant increase in the lattice parameter $a$ and lattice volume $V$ at the end of the discharge. In contrast, there is an obvious minor increase at the end of the discharge for 120, corresponding to the extra capacity (additional peak in the dQ/dV profile) and oxygen redox. Gao et al. recently reported the oxygen redox in the W-doped LiNiO₂ cathode[23]. Similar to some Li-rich materials, a redox inversion behavior exists, the Ni oxidation and oxygen oxidation on charge are followed by a sequential nickel reduction and oxygen reduction on discharge[40]. As a result of such an inversion, there is an extra discharge capacity at the end of discharge, corresponding to the contribution of oxygen reduction[23,40,41]. Owing to the sluggish kinetics of oxygen redox, a voltage hysteresis behavior can be obtained in the voltage profiles[42]. The above behaviors, including extra capacity at lower discharging voltages and voltage hysteresis, are also clearly observed in our cathode materials. Such behaviors become more obvious when the Li content increases from 108 to 120. Especially for 120, the incorporation of oxygen redox greatly alleviates the changes in lattice parameters and lattice volume. Compared to Li-rich materials, the anionic redox activity is minor and it is highly coupled with the cationic redox. The oxygen redox generally causes local structural variation (minor changes in $a$) rather than collective Ni-O bond length change (Ni-redox and lattice changes in $a$)[40]. That is why the signs of the O redox can only be obtained from the lattice parameters at the end of discharge, where the Ni reduction has been almost finished. When the discharge cutoff voltage extends from 2.5 V to 2.0 V, more oxidized O species can be reduced owing to their kinetically difficult nature, and a more complete oxygen reduction process helps increase the cycling stability of the cathode materials (in line with the observation in the electrochemical performance)[40,41]. Gao et al. believe that the oxygen redox is contributed by the W/Ni(Li) ordering domains in the LiNiO₂ lattice. The Li$_{sole}$ (refinement results) and Li$_{sole}$ with Mn neighbors (peak B in NMR) in our material system would have similar structures when they

surround Mo/Nb/W in the same plane. Notice that anionic redox is also active in $d^0/d^{10}$-M-based lithium-rich oxides doped with Ni²⁺, which exactly corresponds to the Ni²⁺-doped Li$_a$XO$_b$ domains for our materials (peak D in NMR)[43–47]. Therefore, it is likely that these two structural units collectively activate the oxygen redox for our materials. A discussion on the final charged and discharged structure for 108, 114, and 120 by refinement can be found in Supplementary Note 5.

In situ XAS studies of the 108, 114, and 120 positive electrodes were conducted to monitor the changes in the valence states of Ni and Mn during charge and discharge (Figs. S16 and S17). The selected XANES results show that the Ni K-edge gradually shifted to higher energy during charge and back during discharge, indicating the Ni oxidation and reduction for all three samples. For the Mn K-edge, some changes might be related to changes in the local structure rather than changes in the oxidation states. Compared to 108, the Ni K-edge of 114 and 120 exhibited a gentle shift to lower energy at the end of discharge, which may indicate the participation of the oxygen redox. This behavior is further confirmed by checking all the in situ spectra during discharge for sample 120 (Fig.S18). The speed of Ni reduction obviously slows down at the end of discharge, indicating the oxygen redox mechanism (in line with the observation of corresponding lattice changes).

To further explain the obvious difference in capacity retention among 108, 114, and 120, ex situ hard XAS was performed at the Ni K-edge on pristine and cycled electrodes (2.5 V–4.4 V, after 50 cycles) for these three samples (Fig. S19). Interestingly, the average valence states of Ni are reduced (left shift) for all three samples during cycling. Normally, there is an oxidation (right shift) for LiNiO₂ and Ni-rich positive electrodes during cycling because Li ions diffusion and a full Ni reduction are impeded owing to the surface rock salt phase[24,48,49]. 108 and 114 show a similar degree of Ni-reduction, whereas the degree almost doubles for 120. The dQ/dV profiles of these three electrodes after certain cycles are also provided here for discussion (Fig. S20). On discharge, extra capacities are obtained below around 3.4 V owing to

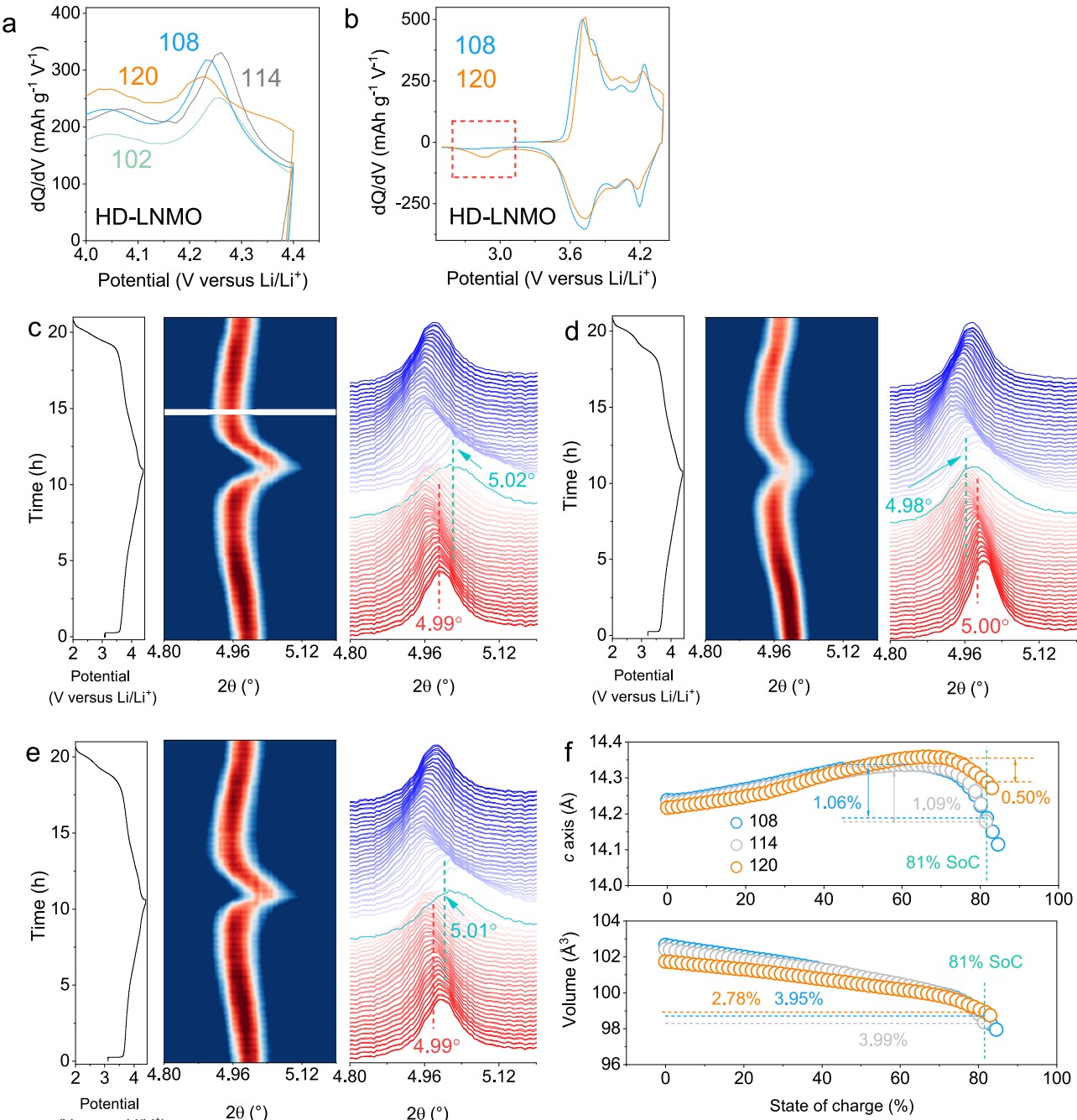

**Fig. 5 | Bulk phase structure stability characterization. a** The enlarged Fig.4b, i.e. dQ/dV profile of HD-LNMO positive electrodes, highlighting the changes in the peak intensity at 4.25 V. **b** The full comparison of the dQ/dV curves between 108 and 120. A clear difference is displayed in the marked area at lower voltage ranges. 2D contour pattern of the in situ SXRD data of (**c**) 108, (**d**) 120, and (**e**) 114 positive electrodes. The in situ cells (with an active material mass loading of around 4 mg cm$^{-2}$) are measured at room temperature (25 °C) and 21 mA g$^{-1}$ in the voltage range of 2.0–4.4 V. The corresponding SXRD patterns are displayed on the right side ($\lambda$ = 0.4139 Å). The location of the 003 reflections in the pristine state and SoC 81% are labeled. **f** $c$-lattice parameter and lattice volume changes of HD-LNMO-108, 120, and 114 positive electrodes during charge.

the oxygen redox, corresponding to the extremely flat peak in the dQ/dV profiles. Such peaks become much more obvious for the profile of 120, in line with the strongest oxygen redox activity. As the cycling proceeds, the intensity of these peaks gradually decreases for all three electrodes, suggesting a reduced oxygen redox. In contrast, an increase in intensity is found in the voltage range of 3.2 V–3.6 V on discharge. This may correspond to the slight reduction of Ni observed in ex situ XAS during cycling. The reduction provides extra redox Ni$^{2+}$/Ni$^{3+}$ and may compensate for the capacity loss during cycling. Such

over-reduction beyond the pristine state may originate from the sluggish kinetics of oxygen on discharge or the irreversible oxygen redox between oxidation and reduction (minor oxygen release). Similar capacity compensation behaviors are also widely observed in the traditional Li-rich systems[50]. Further detailed investigations in future research are needed to probe the complex mechanisms behind the compensation. The oxygen redox as well as the capacity compensation certainly play an important role in the improved performance of the 108 and 120 positive electrodes. Note that the rapid

decay in dQ/dV profiles of 114 suggests a common bulk fatigue behavior for Ni-rich materials (owing to less Li/Ni exchange compared to 108 and less oxygen redox compared to 120): the abrupt anisotropic contraction in the lattice causes mechanical damage (cracking and pulverization), which intensifies the surface irreversible phase transition[51]. The evaluation of the actual mechanical damage only by the lattice changes is not sufficient. The cross-section scanning electron microscope (SEM) images of 108 and 114 positive electrodes after 50 cycles are compared (Fig. S21). There were numerous microcracks with a relatively wide opening in the cycled cathode of 114, while the microcracks in the cycled cathode of 108 were sparsely visible. The microcracks had extended to the particle surface for 114, providing channels for the infiltration of the electrolyte into the inner part. The increased mechanical stabilities of 108 and 120 originate from different reasons: the moderate Li/Ni mixing for 108 helps decrease the abrupt lattice contraction on charge; Although a much smaller mixing is found, the increased oxygen redox for 120 plays the same role in alleviation of the contraction.

In summary, we thoroughly probe different Li occupancies in a Ni-rich layered structure by a combination of Li-sensitive characterization methods. Li exists in various domains and defects, including $Li_aXO_b$, Li/Mn/X(Ni) ordering domains. Based on the charge-balancing principle, a Li-regulation strategy was developed to tune the relative content of each component further. Through this approach, mitigated lattice variation and persistent structural stability in Ni-rich positive electrodes can be realized in two different manners: optimizing the amount of Li/Ni exchange units or introducing an extra oxygen redox. As a consequence, those two stabilized materials exhibit improved electrochemical performance (in comparison to untuned positive electrodes). We believe such research will draw attention to the full investigation of the Li occupation in such Ni-rich positive electrodes. The Li-tuning method can broaden the perspectives for developing long-life positive electrodes for advanced Li-ion battery systems.

## Methods

### Synthesis of HD-LNMO and RM

First, the co-precipitation method was used to prepare the precursors of $Ni_{0.83}Mn_{0.13}Mo_{0.02}Nb_{0.01}W_{0.01}(OH)_2$. A continuous stirred tank reactor (CSTR) with a capacity of 1.2 L was used under a $N_2$ atmosphere. 200 mL of deionized water was added to the CSTR and kept at 60 °C. Three kinds of water solutions (denoted as solution A, solution B, and solution C) were used. Solution A (200 mL) was prepared by dissolving $NiSO_4 \cdot 6H_2O$ (≥98%, Sigma-Aldrich), $MnSO_4 \cdot H_2O$ (≥98%, Sigma-Aldrich), $(NH_4)_6Mo_7O_{24} \cdot 4H_2O$ (≥99.98%, Sigma-Aldrich), and $C_4H_4NNbO_9$ (≥99.99%, Sigma-Aldrich) in deionized water with a total concentration of 1.98 M. Solution B (200 mL) was obtained by dissolving 0.02 M $WO_3$ (≥99.9%, Sigma-Aldrich) in 2 M NaOH (≥98%, Sigma-Aldrich) and 0.6 M $NH_4OH$ (28.0-30.0% $NH_3$ basis, Sigma-Aldrich) mixed water solution. Solution C was 3.0 M NaOH (100 mL). Solution A and solution B were added into the CSTR with the same flow rate simultaneously. The pH of the mixed solution in the CSTR was maintained at $11.0 \pm 0.2$ by controlling the flow rate of solution C. The stirring speed (500 rpm) and temperature of the solution in the CSTR (60 °C) were controlled strictly. The precursor (sediment) was filtered and then washed with deionized water several times until the pH of the filtrate was close to 7.0. The filtered powder was dried at 120 °C overnight. The precursor of RM was synthesized using a similar method, where $CoSO_4 \cdot 7H_2O$ (≥98%, Sigma-Aldrich) was dissolved in solution B.

Second, the dried hydroxide precursor powder is mixed thoroughly with $LiOH \cdot H_2O$ (99.95%, Sigma-Aldrich) powder (for $^6Li$-enriched samples, $^6LiOH \cdot H_2O$ (95% atom% $^6Li$, Sigma-Aldrich) was used) with 2%, 8%, 14%, and 20% excess (molar ratio) Li. The mixtures are calcined in a tube furnace at 550 °C for 6 h and then at 730 °C for 12 h under an oxygen flow of 50 mL $min^{-1}$. Regarding RM, an 8% Li

excess is used and the other conditions for synthesis are the same as described above.

### Diffraction techniques

Synchrotron X-ray diffraction (SXRD) measurements were conducted at beamline P02.1, located at DESY (Hamburg, Germany) with a wavelength of 0.2073 Å. The samples were loaded into a capillary with a diameter of 0.5 mm, and the collecting time for each pattern was 120 s. Neutron diffraction (ND) data were acquired on the D2B high-resolution powder diffractometer of the Institute Laue-Langevin (Grenoble, France), at a wavelength of 1.594 Å. Samples were loaded into cylindrical vanadium cans. Combined Rietveld refinement against the SXRD and ND patterns was performed using Fullprof software. Synchrotron-based in situ XRD experiments were performed at the MSPD beamline at ALBA synchrotron (Barcelona, Spain) with a photon energy of 30 keV ($\lambda = 0.4130$ Å)[52]. The diffraction patterns were acquired using a MYTHEN 2 position-sensitive detector. The exposure time for each pattern was 60 s. CR2025 coin cells with glass windows (diameter = 5 mm) were used.

### Electrode fabrication and electrochemical testing

The electrochemical performances are tested in the CR2025 coin cell. First, the positive electrode slurry is prepared by uniformly mixing the active material, C65 carbon black (MTI Co., Ltd.), and polyvinylidene fluoride (PVDF, Sigma-Aldrich) at a mass ratio of 8:1:1 in a planetary mixer (THINKY ARV-310) with 2000 rpm for 10 min under an air atmosphere and room temperature to ensure homogeneity. The N-methyl pyrrolidone (NMP) with a moisture content of less than 0.1% (VMR) was used as the solvent. Second, the well-mixed slurry is coated on Al foil (15 μm thickness, häberle LABORTECHNIK GmbH & Co.KG) using a ZUA 2000 Universal applicator with a wet thickness of 150 μm. The coated aluminum foil was subsequently dried in an oven at 80 °C under ambient conditions to evaporate the NMP solvent. After a drying period of 6 h, the electrode is cut into discs with a diameter of 12 mm using a handheld punch (NOGAMIGIKEN Co., Ltd.), and the mass loading of active material is 4–5 mg $cm^{-2}$. A subsequent drying step was conducted using a Büchi glass oven (B-585) under vacuum conditions at 120 °C for 12 h. No calendering procedure was used. Finally, the coin cell is assembled in the Ar glovebox using a Li metal chip (14 mm diameter, 0.25 mm thickness, PI-KEM) as a counter electrode and Celgard-2025 (16 mm diameter, 25 μm thickness, 55% porosity, DODO Co., Ltd.) as the separator. The electrolyte is 100 μL 1 M $LiPF_6$ dissolved in a mixture of ethylene carbonate (EC), ethyl methyl carbonate (EMC), and diethyl carbonate (DEC) with a volume ratio of 1:1:1 with moisture content maintained below 10 ppm (DODO Co., Ltd.). All assembly procedures were conducted within an argon-filled glovebox ($O_2$ and $H_2O < 0.1$ ppm). All cells were tested using a VMP3 multichannel potentiostat (Bio-Logic, France) at room temperature (25 °C). To ensure reproducibility, all electrochemical experiments were performed using a minimum of two-coin cells. The current density for each cell was calculated based on the mass of the active material in the electrode. The coulombic efficiency was determined as the percentage ratio of discharging capacity to charging capacity, multiplied by 100. The capacity retention is defined as the specific discharge capacity in the last cycle divided by the largest specific discharge capacity during the cycling. State of charge (SoC) is defined as the $x$ in $Li_{(1-x)}Ni_{0.83}Mn_{0.13}Mo_{0.02}Nb_{0.01}W_{0.01}O_2$ in percentage form. 100% SoC corresponds to the ideal capacity of the positive electrode, i.e. 274 mAh $g^{-1}$. The same electrolyte (also the same usage amount), electrodes, and separator were used for both the in situ cells (including for the in situ SXRD and XAS experiments) and the normal coin cells.

### Solid-state nuclear magnetic resonance (NMR) spectroscopy

All NMR experiments were conducted on a Bruker Avance 200 MHz spectrometer at a magnetic field of 4.7 T. For the $^6Li$ magic-angle

spinning (MAS) NMR experiments, the spectra were acquired with 1.3 mm rotors at a spinning speed of 55 kHz. The recycle delay was set to 1 s and the Larmor frequency was 29.5 MHz. $^6$Li MAS NMR spectra were measured using a rotor-synchronized Hahn-echo pulse sequence (90°-τ-180°-τ-acquisition) with a 90° pulse length of 1.6 μs. The $^6$Li NMR shifts were referenced using an aqueous 1 M $^6$LiCl solution (0 ppm). All spectra were normalized to sample mass and number of scans. The spectra were fitted with the dmfit software, using a Gausso-Lorentzian mixed model[53]. The relative peak ratio was determined by the area of each peak after the fitting.

## STEM, XAS, and ICP-OES experiments

High-angle annular dark field (HAADF) scanning transmission electron microscopy (STEM) analysis was performed using a double-corrected Spectra 300 X-EFG microscope from Thermo Fischer Scientific. This instrument is equipped with a probe and image correctors and was operated at an accelerating voltage of 200–300 kV. HAADF STEM images, captured within an angular range of 72–352 mrad, were acquired with a beam current of ~100 pA. X-ray absorption spectroscopy (XAS) was performed at the XAS beamline of the KIT Light Source (Karlsruhe) and the ID26 of the European Synchrotron Radiation Facility (ESRF). XAS data on pristine materials and the in situ cells (CR2025 coin cells with Kaption windows (diameter = 5 mm) were used) were recorded at the Ni and Mn K-edge in transmission mode. For the XAS data on cycled electrodes, the coin cells were cycled to specific cycles before being disassembled in the glovebox. The obtained electrodes were then thoroughly washed with DMC solvent three times and sealed under vacuum conditions within the glovebox. Background subtraction and normalization were done with the ATHENA software, which is part of the Demeter package. The energy calibration was checked by Ni and Mn metal foils, which were placed between the second and third ionization chambers. The Li, Ni, Co, Mn, Mo, Nb, and W content in the materials was determined via inductively coupled plasma-optical emission spectroscopy (ICP-OES) using a Thermo Fischer Scientific iCAP 7600 DUO. The mass fraction was determined from three independent measurements. About 10 mg of the samples were dissolved in 2 mL of hydrochloric acid, 6 mL of nitric acid, and 2 mL of hydrofluoric acid at 353 K for 4 h in a graphite oven.

## Data availability

Source data are provided as a Source Data file. All data supporting the findings in the study are presented within the main text and the supplementary information. Source data are provided with this paper.

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

## Acknowledgements

The authors gratefully acknowledge Valeriu Mereacre for helping with the material preparation and Liuda Mereacre for support in the lab. This work is supported by the China and Germany Postdoctoral Exchange Program (HGF-OCPC) between Peking University and Karlsruhe Institute of Technology (KIT) under program no. ZD2022028-H.Li, the Major Science and Technology Infrastructure Project of Material Genome Big-science Facilities Platform supported by the Municipal Development and Reform Commission of Shenzhen, the Basic and Applied Basic Research Foundation of Guangdong Province (No. 2021B1515130002-F.P.), the Soft Science Research Project of Guangdong Province (No. 2017B030301013-F.P.), International Joint Research Center for Electric Vehicle Power Battery and Materials (No.2015B01015-F.P.), and the Shenzhen Science and Technology Research Grant (No. ZDSYS201707281026184-F.P.). The research used the resources of the P02.1 beamline of the PETRA III at DESY (Hamburg, Germany), a member of the Helmholtz Association. This research used resources from the MSPD beamline at the ALBA synchrotron (Barcelona, Spain). This work contributes to the research performed at CELEST (Center for Electro-chemical Energy Storage Ulm-Karlsruhe). ICN2 acknowledges funding from Generalitat de Catalunya 2021SGR00457-J.A. This study is part of the Advanced Materials programme and was supported by MCIN with funding from European Union NextGenerationEU (PRTR-C17.I1-J.A.) and by Generalitat de Catalunya. The authors thank support from the project NANOGEN (PID2020-116093RB-C43), funded by MCIN/ AEI/10.13039/501100011033/ and by "ERDF A way of making Europe", by the "European Union". ICN2 is supported by the Severo Ochoa program from Spanish MCIN/AEI (Grant No.: CEX2021-001214-S) and is funded by the CERCA Programme/Generalitat de Catalunya. Authors acknowledge the use of instrumentation as well as the technical advice provided by the Joint Electron Microscopy Center at ALBA (JEMCA). ICN2 acknowledges funding from Grant IU16-014206 (METCAM-FIB) funded by the European Union through the European Regional Development Fund (ERDF), with the support of the Ministry of Research and Universities, Generalitat de Catalunya. ICN2 is a founding member of e-DREAM[54]. We acknowledge the European Synchrotron Radiation Facility (ESRF) for the provision of synchrotron radiation facilities and we would like to thank Jing Lin and Dr. Sami Juhani Vasala for assistance and support in using beamline ID26. We acknowledge Bijian Deng for the SEM experiments.

## Author contributions

F.P., H.E., and S.I. directed the research projects and supervised the work. H.Li synthesized the materials, carried out electrochemical measurements, and analyzed the data. E.S. set up the ND experiment and H.Li and S.I. finished the measurement. T. B. conducted the ICP-OES measurement. S.M. conducted the XAS measurement and discussed the results with S.I. and H.Li. V.B. set up the SXRD experiment and H.Liu performed the measurement. A.M. set up the in situ SXRD experiment and S.I., H.Li, and H.Liu performed the experiment. S.L., Y, Z., and J.A. performed the STEM measurement and helped with STEM data analysis. H.Li analyzed the ex situ and in situ SXRD data with the advice given by H.Liu, W.H., and M.K. H.Li conducted the MAS-NMR measurement and discussed it with S.I. H.Li wrote the manuscript with input from all co-authors. All authors commented and reviewed the manuscript.

## Funding

## Competing interests

The authors declare no competing interests.
