## [Transparent Peer Review file · Nature Communications]

Tuning Li Occupancy and Local Structures for Advanced Co-Free Ni-Rich Positive Electrodes

Corresponding Author: Dr Hang Li

Version 0:

Reviewer comments:

Reviewer #1

(Remarks to the Author)

I recommend publication of this sound work after the issues below are addressed:

Abstract and onwards: "all different Li occupancies" is an overreaching statement and needs rewording. There is no all powerful methods and all methods has intrinsic limits, including the XRD/NMR analysis used here has limitations as well and certain assumptions were made when considering Li occupancies. Furthermore, I believe no such claim is necessary for a scientific publication anyways.

Similar wording, such as "all kinds of Li-occupancies", needs to be fixed in conclusions as well. Instead, I think specificity rather than generalization will help to give a clearer and sufficient message in this paper. The methodology essentially informs on 4 types of Li occupancies shown in Figure 3 but the presence of more (and important ones) may be discovered later.

Line 149: Reconsider the word "good" in describing proper layering

Are the 6Li enriched samples identical to the parent natural abundant compositions ? It should be discussed. Did the authors check their (super) lattice structures, impurities etc. ? It is not uncommon to have differences in the synthetic product due to slight changes in precursors due to differences in isotope enriched materials vs natural abundant ones.

Line 179: The basis for the assignment for the 296ppm peak B is not clear. The cited studies show a large hyperfine shift for Li coordinated to Mn⁺⁴, then why is the shift observed for peak B so small, almost by half an order of magnitude ?

These half cells should have additional detrimental effects in cycling performance to be considered vs. Li metal after a certain number of cycles.

The discussion about experimental evidence for a conformal epitaxial layer between the rock salt phase and layered phase is interesting but lacks supporting evidence. This must be discussed appropriately and perhaps suggested as a future direction rather than a conclusion. The only direct evidence is the presence of it in HDAF for 108 but 120 is not shown, while the favorable effect and discussion is for 120 as far as I can tell.

There's an apparent phasing problem in fig 5g charged sample. Also quantification by eye for discharged sample suggests a big loss of Li. Why is it so high ? Needs to be discussed.

Reviewer #2

(Remarks to the Author)

Although various strategies such as doping and coating have been proposed to improve the bulk and surface instability of layered Ni-rich cathodes, the impact of differences in Li occupation within the lattice structure on electrochemical stability

has not been sufficiently studied. In this paper, the authors categorized Li occupation into ordered Li/Ni, disordered Li/Ni, and rock-salt LiaXOb (X = high-valence dopants; Mo⁶⁺, Nb⁵⁺, and W⁶⁺), and analyzed the electrochemical stability according to changes in Li occupation. However, there is a lack of logical connection between the differences in Li occupation and electrochemical stability, and several important issues remain unresolved. Thus, I cannot recommend it for publication.

1. To analyze the changes in Li occupation according to the Li content during the synthesis process and its effects, the authors compared the electrochemical stability of Li_xNi_{0.83}Mn_{0.13}Mo_{0.02}Nb_{0.01}W_{0.01}O₂ (HD-LNMO-102, 108, 114, and 120, where the numbers represent Li content of 1.02, 1.08, 1.14, and 1.20). However, to analyze performance solely based on differences in Li occupation, it would be more effective to use a single dopant. The authors did not explain why they added three elements, Mo, Nb, and W, as high-valence dopants for the formation of LiaXOb. Additionally, as shown in the ICP-OES analysis in Table S1, the composition of Mo, Nb, and W varies among the HD-LNMO samples as the Li content changes. This variation can significantly affect the differences in electrochemical performance among the samples, especially considering the very low amounts of dopants, which raises concerns about the reliability of the electrochemical analysis.

2. In the comparison of electrochemical properties shown in Fig. 4c, the capacity retention of HD-LNMO-120 over 100 cycles is higher in the voltage range of 2.0 V - 4.3 V (~93.5%) than in the range of 2.5 V - 4.3 V (~91.4%). Generally, under a lower cut-off voltage condition, capacity retention is expected to decrease due to more severe anisotropic structural deformation caused by worsened Jahn-Teller distortion and greater changes in Li⁺ content within the lattice. The authors need to explain why the capacity retention is higher under the 2.0 V condition compared to the 2.5 V condition. Additionally, a comparative analysis of cycle performance in the 2.0 V - 4.3 V voltage range should be conducted for the other samples (HD-LNMO-102, 108, 114, and RM) as well.

3. In the in-situ SXR analysis shown in Fig. 5c and Fig. S8, even though the structural deformation during charge/discharge is smaller in HD-LNMO-114 than in HD-LNMO-108, the cycle performance of HD-LNMO-114 is lower. The authors attributed this to lower surface stability caused by less Li/Ni disorder in HD-LNMO-114. However, since no data comparing the surface stability of the two samples is provided, additional analysis is required.

4. In Fig. 5c, the pattern corresponding to ~3.9 V during the discharge process of HD-LNMO-108 is missing, but this is not addressed. The data should need to be remeasured.

5. The authors stated that the phase transition of HD-LNMO-120 was more stable than that of HD-LNMO-114, citing the difference in peak shift corresponding to (003) in the in-situ SXR. However, in Fig. S8 and Fig. 5d, the peak shift corresponding to (003) in HD-LNMO-114 and HD-LNMO-120 appears as 4.99° to 5.01° and 5.00° to 4.98°, respectively. This is insufficient to explain the significant difference in capacity retention during 80 cycles (at 2.5 V - 4.4 V vs. Li⁺/Li) between HD-LNMO-114 (~78.0%) and HD-LNMO-120 (~92.2%). Moreover, in Fig. 4c-d, HD-LNMO-120, which showed the highest cycle performance, experienced the most severe irreversible decrease in peak intensity during the discharge process in the in-situ SXR among all the samples. Therefore, the comparison of structural stability through in-situ SXR does not sufficiently explain the differences in electrochemical stability among the samples, and additional analysis is also needed.

Reviewer #3

(Remarks to the Author)

Co-Free Ni-rich cathodes are investigated to battery properties, crystal structure, valence by NMR, and local structure using quantum beams with varying Li compositions.

The relationship between the crystal structure and the local structure of the battery properties is not clear. It is necessary to reexamine the relationship including other methods before resubmitting.

Some comments are given below.

1. p.5 ICP composition should be considered including error bar.
2. p.7 It is important to examine the valence of Ni and Mn by XANES and other methods to see the differences in composition and changes in the charge-discharge process.
3. p.7 It is also necessary to consider how Mo, Nb, and W substitution affects this system.
4. p.10 Structure analysis including Fm-3m observed by HAADF and FFT in the second phase is also necessary.
5. p.12 Compositional changes and charging/discharging processes should be analyzed and discussed according to bonding distances, polyhedral distortion, BVS, etc. obtained from structural analysis.
6. Fig.3 The structural model needs to be validated and discussed by other methods (EXAFS, PDF).
7. Fig.4 It is necessary to consider the characteristic improvement factor by the discharge cutoff potential.
- Fig.4 a, b Numerical values for vertical axis and caption need to be added.
9. Since the properties of Li composition 1.2 are good, it is necessary to investigate with a higher composition. It is also not clear why 1.2 is better.
10. Table S2 - S6 The unit of R factor needs to be added. %

Version 1:

Reviewer comments:

Reviewer #1

(Remarks to the Author)

The revisions made are satisfactory for publication.

Reviewer #2

(Remarks to the Author)

All reviewers' concerns have been satisfactorily addressed. Therefore, I recommend accepting this manuscript for publication.

Reviewer #3

(Remarks to the Author)

The author has responded approximately to the revisions noted with corrections and additions.

Point-to-point Response to Reviewers' Comments

Tuning Li Occupancy and Local Structures for Advanced Co-Free Ni-Rich Cathodes

We sincerely thank the editor and all reviewers for their valuable review comments that we have used to improve the quality of our manuscript. The reviewer comments are laid out below and specific concerns have been numbered, and all the responses are listed in the following section, point by point.

Reviewer # 1 (Remarks to the Author):

I recommend publication of this sound work after the issues below are addressed:

Our reply: Thank you for the high evaluation of our work. In the following, we will address all the comments sequentially.

Comment # 1: Abstract and onwards: “all different Li occupancies” is an overreaching statement and needs rewording. There is no all powerful methods and all methods has intrinsic limits, including the XRD/NMR analysis used here has limitations as well and certain assumptions were made when considering Li occupancies. Furthermore, I believe no such claim is necessary for a scientific publication anyways. Similar wording, such as “all kinds of Li-occupancies”, needs to be fixed in conclusions as well. Instead, I think specificity rather than generalization will help to give a clearer and sufficient message in this paper. The methodology essentially informs on 4 types of Li occupancies shown in Figure 3 but the presence of more (and important ones) may be discovered later.

Our reply: Thanks for the suggestion. I fully agree with your opinion. The sentences in the abstract and the conclusions are changed. Instead of ‘all different/kinds of Li occupancies’, the specific Li-containing structural units (Ni/Li exchange, Li_aXO_b , Li/Mn/X(Ni) ordering domains, X= Nb^{5+} , W^{6+} , and Mo^{6+}) are used to show what we investigate in this research. Please see the corresponding changes on page 2 and page 23 (highlighted version).

Comment # 2: Line 149: Reconsider the word “good” in describing proper layering

Our reply: Thanks for the suggestion. The phrase ‘good layering’ is replaced by ‘cation ordering’ to be more specific and proper. Please see the changes on page 6.

Comment # 3: Are the ^6Li enriched samples identical to the parent natural abundant compositions? It should be discussed. Did the authors check their (super) lattice structures, impurities etc.? It is not uncommon to have differences in the synthetic product due to slight changes in precursors due to differences in isotope enriched materials vs natural abundant ones.

Our reply: Thanks for the good suggestion. Powder X-ray diffraction experiments (see the graph below) confirm that all ^6Li enriched samples have a NaFeO_2 -type layered structure. No other peaks are observed, confirming the high purity of the materials. ^7Li NMR of normal samples (without ^6Li enriched) are shown below. The same features as ^6Li NMR spectra can be obtained, although the spinning sidebands appear: the intensity of peak B and peak D grow when Li content is increased. Such consistency proves that ^6Li enriched and non-enriched samples have the same structure and local environments. These discussions are added on page 9 and the supporting information in Note 3.

Comment # 4: Line 179: The basis for the assignment for the 296ppm peak B is not clear. The cited studies show a large hyperfine shift for Li coordinated to Mn⁴⁺, then why is the shift observed for peak B so small, almost by half an order of magnitude?

Our reply: Thanks for the question. The difference in the NMR shift originates from the different Li local environments owing to the different material systems. The cited study (<https://iopscience.iop.org/article/10.1149/1.1737711/meta>) investigates the structure of the Li-rich Mn-rich materials, where Li in the TM layers is surrounded by six or five Mn⁴⁺ in the same plane (the so-called honeycomb structure). In contrast, owing to the Ni-rich Mn-poor nature of our materials, Li in the TM layers may only be surrounded by one Mn⁴⁺. Other neighboring atoms surrounding Li include Ni, Mo, Nb, and W. If the NMR shift induced by the six Mn⁴⁺-O-Li interactions in the study is used, one Mn⁴⁺-O-Li interaction can induce an NMR shift of 290 ppm. Since Ni, Mo, Nb, and W only induce minor NMR shifts, the NMR signal of Li in TM layers in our material mainly results from one Mn⁴⁺-O-Li interaction. The calculated value (290 ppm) is close to the NMR shift of peak B (296 ppm), suggesting the consistency between the cited study and our work regarding the value of the single Mn⁴⁺-O-Li interaction. The above discussion has been added to the corresponding part to clearly illustrate the ownership of peak B on page 9.

Comment # 5: These half cells should have additional detrimental effects in cycling performance to be considered vs. Li metal after a certain number of cycles.

Our reply: Thanks for the question. Several issues in the half cells (excluding the cathode part) can influence the cycle life and cycling performance. First, the continuous formation of thick, heterogeneous, and porous SEI on the Li metal causes both electrolyte and Li metal depletion <https://www.nature.com/articles/s41560-019-0338-x>; Second, the exposure of freshly deposited Li during repeating Li plating processes induces further electrolyte consumption; Third, the formation of Li dendrite leads to possible micro-shorts. When the half cells are assembled, excess electrolytes and thick Li anodes are used so the above detrimental issues have minor effects on the cell performance before around 100 cycles <https://www.nature.com/articles/s41467-021-26815-6>. The evaluation of long-term performance (more than 100 cycles) should be based on the full cell configuration. However, the half-cell configuration allows scientists to focus on the new materials (not considering the N/P ratio, etc.) and conduct an accurate assessment of them, including the voltage curves, capacities, and rate capability. One sentence was added on page 16 to inform readers that the above issues may cause extra performance decay during cycling.

Comment # 6: The discussion about experimental evidence for a conformal epitaxial layer between the rock salt phase and layered phase is interesting but lacks supporting evidence. This must be discussed appropriately and perhaps suggested as a future direction rather than a conclusion. The only direct evidence is the presence of it in HDAF for 108 but 120 is not shown, while the favorable effect and discussion is for 120 as far as I can tell.

Our reply: Thanks for the suggestion. Many researchers have reported the formation of the Li_aXO_b (X could be Mo, Nb, W, Ta) phase on particle surfaces when high-valence-state dopants are used in Ni-rich materials <https://doi.org/10.1039/D2EE03969A>, <https://10.1002/aenm.202103067>. Depending on Li and X's content, such phases' exact structure may vary. As requested, the STEM experiment was conducted on the 120 sample, and the corresponding HAADF image is shown below. Similar to the image of 108, an obvious decrease in the contrast of the TM layer is observed on the left side, indicating the replacement of Ni and Mn by Li. Two regions are marked (A and B) and both can be indexed to a layered structure ($R\bar{3}m$), as verified by the FFT patterns from the selected areas. However, the two domains exhibit different characteristic periodic atomic arrangements in the Li layers, as shown by the height profiles of lines 1 and 2. The contrast in the Li layers has been greatly increased from region A to region B, suggesting a higher Li/Ni exchange in region B. Why an $R\bar{3}m$ structure can be kept in the 120 but not in the 108? Our understanding is that Ni^{2+} may play an important role in phase transformation. The 120 sample has less Ni^{2+} (obvious peak splitting and less Li/Ni mixing), thus a larger tolerance for the atomic rearrangement in a single phase. Such a finding also proves that the epitaxial growth of the two-phase is not the reason for the high stability of the cathodes. The introduction of the oxygen redox by the Li/Mn/X(Ni) ordering domains should be the main reason for excellent performance in the 120 cathode. Therefore, in the revised manuscript, the focus has been transferred from the epitaxial layer to the Li/Ni exchange and the oxygen redox. The influence of the oxygen redox on the stability of the cathodes should be the key point for further future research. The corresponding discussion has been added in the main text on page 13 and Supplementary Note 4.

Comment # 7: There's an apparent phasing problem in fig 5g charged sample. Also quantification by eye for discharged sample suggests a big loss of Li. Why is it so high? Needs to be discussed.

Our reply: Thanks for the questions. For the spectra of the charged sample, there is no phasing problem. The right shift of the NMR signal is induced by the removal of Li and the related oxidation of Ni <https://pubs.acs.org/doi/10.1021/acs.chemmater.9b00140>. The NMR signal would keep moving to lower ppm positions on continuous Li removal since Ni³⁺ ions keep being oxidized to Ni⁴⁺, and Ni⁴⁺ is diamagnetic (low-spin configuration (3d⁶) in an octahedral environment). Therefore, the original contribution from the Ni³⁺-O-Li interaction in the large NMR shift will be removed.

Regarding the spectrum of the discharged sample, the loss of ⁶Li after discharge is caused by the insertion of the ⁷Li from the electrolyte and the Li anode during discharge. As displayed below, the Coulomb efficiency of the 1st cycle for the measured sample is 96.2%, which means there is only a Li loss of 3.8%. However, some of the reinserted Li ions are ⁷Li rather than ⁶Li so they cannot be detected by the ⁶Li NMR. Another issue is that for charged and discharged samples, the active material only accounts for 80% (mass%) in the electrode. In contrast, this number is 100% for the pristine sample (powder). Therefore, the intensity of the spectra for the charged and discharged samples should be increased 1.25 times for a comparison with that of pristine.

Reviewer # 2 (Remarks to the Author):

Although various strategies such as doping and coating have been proposed to improve the bulk and surface instability of layered Ni-rich cathodes, the impact of differences in Li occupation within the lattice structure on electrochemical stability has not been sufficiently studied. In this paper, the authors categorized Li occupation into ordered Li/Ni, disordered Li/Ni, and rock-salt Li_xXOb ($X = \text{high-valence dopants; Mo}^{6+}, \text{Nb}^{5+}, \text{and W}^{6+}$), and analyzed the electrochemical stability according to changes in Li occupation. However, there is a lack of logical connection between the differences in Li occupation and electrochemical stability, and several important issues remain unresolved. Thus, I cannot recommend it for publication.

Our reply: Thank you for your careful and in-depth review of our manuscript. In the following, we will address all the comments and revise the manuscript accordingly.

Comment # 1: To analyze the changes in Li occupation according to the Li content during the synthesis process and its effects, the authors compared the electrochemical stability of $\text{Li}_x\text{Ni}_{0.83}\text{Mn}_{0.13}\text{Mo}_{0.02}\text{Nb}_{0.01}\text{W}_{0.01}\text{O}_2$ (HD-LNMO-102, 108, 114, and 120, where the numbers represent Li content of 1.02, 1.08, 1.14, and 1.20). However, to analyze performance solely based on differences in Li occupation, it would be more effective to use a single dopant. The authors did not explain why they added three elements, Mo, Nb, and W, as high-valence dopants for the formation of Li_xXOb . Additionally, as shown in the ICP-OES analysis in Table S1, the composition of Mo, Nb, and W varies among the HD-LNMO samples as the Li content changes. This variation can significantly affect the differences in electrochemical performance among the samples, especially considering the very low amounts of dopants, which raises concerns about the reliability of the electrochemical analysis.

Our reply: One of the purposes of using multiple dopants is to increase the total content of dopants in the Ni-rich cathodes, so a larger change in Li occupation in the lattice will be expected. When using a single dopant, it is challenging to increase the doping content, especially for the high-valence-state dopants, since they are easier to accumulate on the particle surface and form secondary phases. If multiple dopants are applied, the doping content for every single dopant is tiny (1-2%), but the total content can reach a higher value (5-7%). That is why impurity phases can still be found by X-ray diffraction when the doping contents are just 2% for Nb or W. (<https://doi.org/10.1021/acs.chemmater.2c01461> for the Nb doping and <https://doi.org/10.1002/aenm.202103067> for the W doping). The other purpose of using multiple dopants is to acquire the best performance of the Ni-rich cathodes. The combination of multiple dopants can maximize the configurational entropy of the cathode and achieve a robust structure of the cathode. This strategy is borrowed from the entropy stabilization in metallic alloys. The above discussion has been added in the main text (highlighted version) on page 4.

The contents of Mo, Nb, and W are the same for all the materials because these dopants were introduced during the synthesis of the hydroxide precursors, and all the materials were obtained by the calcination of the mixture of Li salts (LiOH) and the precursors. The change in the composition of the ICP-OES results originates from the variation of the Li content. It corresponds to an increase of Li occupation in the TM layers: during the analysis of the ICP-OES results, Li ions are believed to occupy sites in both Li and TM layers, so a formula must be obeyed: $a(\text{Li content}) + b(\text{TM content}) = 2$. When Li content is gradually increased in materials from 102 to 120, more Li ions are located in the TM layers, so the ratio of other TM ions in the TM layers has to be decreased. The ratios of Mn, Nb, Mo, and W are all influenced. However, that decrease doesn't correspond to the changes in the doping content.

Comment # 2: In the comparison of electrochemical properties shown in Fig. 4c, the capacity retention of HD-LNMO-120 over 100 cycles is higher in the voltage range of 2.0 V - 4.3 V (~93.5%) than in the range of 2.5 V - 4.3 V (~91.4%). Generally, under a lower cut-off voltage condition, capacity retention is expected to decrease due to more severe anisotropic structural deformation caused by worsened Jahn-Teller distortion and greater changes in Li⁺ content within the lattice. The authors need to explain why the capacity retention is higher under the 2.0 V condition compared to the 2.5 V condition. Additionally, a comparative analysis of cycle performance in the 2.0 V - 4.3 V voltage range should be conducted for the other samples (HD-LNMO-102, 108, 114, and RM) as well.

Our reply: To answer this question, the redox mechanism of the electrodes is further investigated. In a recent paper, Gao et al. report an oxygen redox in the W-doped LiNiO₂ cathode <https://doi.org/10.1002/aenm.202402793>. Similar to some reported Li-rich materials <https://doi.org/10.1038/s41563-022-01278-2>, a redox inversion behavior exists, the Ni oxidation and oxygen oxidation on charge is followed by a sequential nickel reduction and oxygen reduction on discharge. Specifically, the removal of electrons from σ -type Ni $3d-O2p$ first takes place on charge, corresponding to the oxidation of Ni. The oxidized Ni species then facilitate a charge transfer from O $2p$ non-bonding states to Ni species (so-called LMCT process), which further triggers the anionic redox activity. The LMCT process is associated with the origin of cationic-anionic redox inversion. As a result of such an inversion, there is an extra discharge capacity in the voltage range between 3.4 V and 2.0 V, corresponding to the contribution of oxygen reduction. Owing to the sluggish kinetics of oxygen redox, a voltage hysteresis behavior can be obtained in the voltage profiles.

The above behaviors, including extra capacity at lower discharging voltages and voltage hysteresis, are also clearly observed in our cathode materials. Such behaviors become more obvious when the Li content increases from 108 to 120. Especially for 120, the increased oxygen redox activity can be reflected in the limited changes in lattice parameters and lattice volume on charge, because oxygen redox generally causes O-O

bond shortening rather than collective Ni-O bond length change. Notice that compared to Li-rich materials, the anionic redox activity is minor and it is highly coupled with the cationic redox. That is why the signs of the O redox can only be obtained at the end of discharge, where the Ni reduction has been almost finished. A comparison between 120 and RM electrodes on lattice changes during discharge is shown below. RM, without the oxygen redox, has a constant increase in the lattice parameter a and lattice volume at the end of the discharge. In contrast, there is an obvious minor increase at the end of the discharge for 120, corresponding to the oxygen redox and extra capacity. This behavior is further confirmed by checking all the *in situ* spectra during discharge for sample 120 (see below). The speed of Ni reduction slows down at the end of discharge, indicating the oxygen redox mechanism (in line with the observation of corresponding lattice changes). When the cut-off voltage is changed from 2.5 V to 2.0 V, more oxidized O species can be reduced owing to their kinetically difficult nature, and a more complete oxygen reduction process helps increase the cycling stability of the cathode materials. A study on the battery performance of all samples in the range of 2.0-4.3 V was conducted. The results are shown below. The capacity retention of 102 is improved when the lower cut-off voltage is applied. However, the performance of RM is destroyed, since no oxygen redox exists and a lower cut-off voltage only induces extra side reactions. The corresponding discussion has been added in the 'Structural stability' part on pages 20 and 21, explaining the new discoveries in the redox mechanisms.

Comment # 3: In the in-situ SXRD analysis shown in Fig. 5c and Fig. S8, even though the structural deformation during charge/discharge is smaller in HD-LNMO-114 than in HD-LNMO-108, the cycle performance of HD-LNMO-114 is lower. The authors attributed this to lower surface stability caused by less Li/Ni disorder in HD-LNMO-114. However, since no data comparing the surface stability of the two samples is provided, additional analysis is required.

Our reply: That is a good question. In the manuscript, the large performance difference between 108 and 114 is not fully discussed. Although the reflection peak shifting is similar for 108 (5.02°) and 114 (5.01°) in the *in situ* XRD patterns, the changes in the *c*-axis lattice parameter (Δc) and the lattice volume (ΔV) are different. As shown below, the maximum lattice parameter contraction (Δc) increases from 1.06% for 108 to 1.09% for 114. The same increasing trend is also observed for ΔV : 3.95% for 108 and 3.99% for 114. The dQ/dV profiles after every 15 cycles are plotted and displayed below. The peak at 4.1 V mainly corresponds to the hexagonal to hexagonal ($H2 \rightarrow H3$) phase transition. Such peaks remained reversible during cycling for 108. In contrast, the peaks are dramatically broadened and shifted, and the intensity has also considerably decreased during cycling. Such fading behavior is generally associated with surface irreversible phase transition to NiO-like rock salt phase, which is aggravated by the cracking and pulverization of the electrodes <https://pubs.acs.org/doi/10.1021/acsami.9b09754>. The abrupt contraction in the lattice parameter and lattice volume causes microcracking and fracture of the electrode (see the cross-section SEM image of cycled electrodes of 108 (a and b) and 114 (c and d) below). The growing cracks expose new surfaces to the electrolyte and a new rock salt phase can be gradually formed between the electrolyte and the reactive oxidized Ni species <https://pubs.acs.org/doi/10.1021/acs.chemmater.7b05269>. Such inactive phases increase polarization (the peak shifting in dQ/dV curves of 114) and hinder Li ion migration (decreased peak intensity). The smaller extent of Li/Ni mixing in 114 (4.9%) than that of 108 (6.4%) causes severe contraction in lattice parameters and volume, inducing increased surface-side reactions. An interesting question may arise: why the 120 sample still has a good electrochemical performance, although an even smaller value of Li/Ni exchange (3.2%) is found than that of 114? As explained in the last question, there is an increased oxygen redox activity in 120, which helps decrease the changes in the lattice on charge and discharge. Therefore, the cycling stability of the 120 is less influenced by the Li/Ni exchange. In conclusion, 114 has much-decreased bulk stability owing to less Li/Ni exchange compared to 108 and less oxygen redox compared to 120. The corresponding discussion has been added on pages 20 and 22.

Comment # 4: In Fig. 5c, the pattern corresponding to ~3.9 V during the discharge process of HD-LNMO-108 is missing, but this is not addressed. The data should need to be remeasured.

Our reply: The pattern (78th, discharge voltage 3.885 V) was not shown because the background is different from all other patterns owing to the slight changes in the hardware, such as the short-term intensity changes in the beam. The same behavior was also found in the 62th pattern (discharge voltage 4.178 V) for the RM material. During our *in situ* measurement, eight cells are measured one after another. There were only two patterns that had such a change in the background intensity. After that short time, the background turned back to the original intensity. As shown below, for those two patterns, there is no new peak observed and no sudden changes in the peak positions. The *in situ* cells were not influenced by this “short accident”, as evidenced by the corresponding potential-time curves of the batteries. Therefore, the *in situ* data are trustworthy and can be used for the analysis (especially in the text, we mainly focus on the charging period, and both of these two patterns correspond to the structure on discharge).

Comment # 5: The authors stated that the phase transition of HD-LNMO-120 was more stable than that of HD-LNMO-114, citing the difference in peak shift corresponding to (0 03) in the in-situ SXR. However, in Fig. S8 and Fig. 5d, the peak shift corresponding to (003) in HD-LNMO-114 and HD-LNMO-120 appears as 4.99° to 5.01° and 5.00° to 4.98° , respectively. This is insufficient to explain the significant difference in capacity retention during 80 cycles (at 2.5 V - 4.4 V vs. Li+/Li) between HD-LNMO-114 (~78.0%) and HD-LNMO-120 (~92.2%). Moreover, in Fig. 4c-d, HD-LNMO-120, which showed the highest cycle performance, experienced the most severe irreversible decrease in peak intensity during the discharge process in the in-situ SXR among all the samples. Therefore, the comparison of structural stability through in-situ SXR does not sufficiently explain the differences in electrochemical stability among the samples, and additional analysis is also needed.

Our reply: Thanks for the question. Similar to question 3, the peak shifting cannot fully reflect the lattice change here. The changes in the lattice parameter c (Δc) are 1.09% for 114 and 0.50% for 120. That is a huge difference. The changes in the lattice volume (ΔV) are 3.99% for 114 and 2.78% for 120. Such differences have a profound effect on structure fatigue and electrode aging. To further explain the obvious difference in capacity retention among 108, 114, and 120, *ex situ* hard XAS was performed at the Ni K-edge on pristine and cycled electrodes (after 50 cycles) for these three samples. Interestingly, the average valence states of Ni are reduced (left shift) for all three samples during cycling. Normally, there is an oxidation (right shift) for LiNiO_2 and Ni-rich cathodes during cycling because Li ions diffusion and a full Ni reduction are impeded owing to the surface rock salt phase <https://pubs.acs.org/doi/10.1021/acsami.6b11111>. 108 and 114 show a similar degree of Ni-reduction, whereas the degree almost doubles for 120. The dQ/dV profiles of these three electrodes after certain cycles are also provided here for discussion. On discharge, extra capacities are obtained below around 3.4 V owing to the oxygen redox,

corresponding to the extremely flat peak in the dQ/dV profiles. Such peaks become much more obvious for the profile of 120, in line with the strongest oxygen redox activity. As the cycling proceeds, the intensity of these peaks gradually decreases for all three electrodes, suggesting a reduced oxygen redox. In contrast, an increase in intensity is found in the voltage range of 3.2 V to 3.6 V on discharge. This may correspond to the slight reduction of Ni observed in *ex situ* XAS during cycling. The reduction provides extra redox Ni^{2+}/Ni^{3+} and may compensate for the capacity loss during cycling. Such over-reduction beyond the pristine state may originate from the sluggish kinetics of oxygen on discharge or the irreversible oxygen redox between oxidation and reduction (minor oxygen release). Similar capacity compensation behaviors are also widely observed in the traditional Li-rich systems <https://doi.org/10.1038/s41560-018-0207-z>. Further detailed investigations are needed to probe the complex mechanisms behind the compensation. The oxygen redox as well as the capacity compensation certainly play an important role in the excellent performance of the 108 and 120 cathodes. The above discussion has been added to the 'Structural stability' part on page 21 and 22.

Reviewer # 3 (Remarks to the Author):

Co-Free Ni-rich cathodes are investigated to battery properties, crystal structure, valence by NMR, and local structure using quantum beams with varying Li compositions. The relationship between the crystal structure and the local structure of the battery properties is not clear. It is necessary to reexamine the relationship including other methods before resubmitting. Some comments are given below.

Our reply: Thank you for the careful and in-depth review of our manuscript. We performed several new experiments to further investigate the relationship between the structure and the electrochemical behaviors. We will address all the comments and revise the manuscript accordingly.

Comment # 1: p.5 l.8 ICP composition should be considered including error bar.

Our reply: The error bars are added to the ICP table in the supporting information Table S1.

Comment # 2: p.7 It is important to examine the valence of Ni and Mn by XANES and other methods to see the differences in composition and changes in the charge-discharge process.

Our reply: *In situ* X-ray absorption spectroscopy (XAS) studies of the 108, 114, and 120 cathodes were conducted to monitor the changes in the valence states of Ni and Mn during charge and discharge. The X-ray absorption near edge structure (XANES) results show that the Ni K-edge gradually shifted to higher energy during charge and back during discharge, indicating the Ni oxidation and reduction for all three samples. For the Mn K-edge, although there are some changes, it might be related to changes in the local structure rather than changes in the oxidation states. Compared to 108, the Ni K-edge of 114 and 120 exhibited a gentle shift to lower energy at the end of discharge, which may indicate the participation of the oxygen redox. The corresponding discussion has been added in the main text (the highlighted version) on page 21.

Comment # 3: p.7 It is also necessary to consider how Mo, Nb, and W substitution affects this system.

Our reply: Thanks for the suggestion. On page 7, the structures of this system are discussed based on the combined Rietveld refinement against SXRD and ND data. To investigate the influence of Mo, Nb, and W dopants on structure, the reference material (RM) was synthesized, in which conventional Co replaced those high-valence-state dopants. The comparison in structure between RM and HD-LNMO-108 shows that the introduction of Mo, Nb, and W induces a higher degree of Li/Ni mixing since additional Ni²⁺ ions are formed <https://onlinelibrary.wiley.com/doi/full/10.1002/aenm.202402793>. The decreased average oxidation state of Ni in RM (compared to HD-LNMO-108) can also be checked by the corresponding XANES data. The doping of Mo, Nb, and W increases the lattice parameters and decreases the hexagonality (*c/a* ratio). This part has been rewritten to discuss the effects of substitution in more detail on page 8.

Comment # 4: p.10 Structure analysis including Fm-3m observed by HAADF and FFT in the second phase is also necessary.

Our reply: Thanks for the suggestion. In the discussion of the NMR spectra, the Fm-3m phase observed by the STEM corresponds to the peak D at around 4 ppm. This NMR signal belongs to the Li_aXO_b phase. The detailed structure and distribution of this phase are then confirmed by the HAADF and FFT images. Since all the reflection peaks from the Fm-3m phase will be overlapped by the stronger reflections from the bulk R-3m phase at the same positions, only a single-phase model is used during the combined refinement for the sake of simplicity and repeatability. The Li_{sole} given by the refinement could serve as a whole description for the Li_aXO_b phase as well as the Li in TM layers surrounded by one Mn⁴⁺ neighbor. The corresponding part has been rewritten to provide a more detailed discussion on page 14.

Comment # 5: p.12 Compositional changes and charging/discharging processes should be analyzed and discussed according to bonding distances, polyhedral distortion, BVS, etc. obtained from structural analysis.

Our reply: Since there is an obvious difference in electrochemical performance between 108, 114, and 120, a detailed structural analysis was conducted based on the in situ XRD patterns. For each material, three patterns were chosen (pristine, charged states (81% SoC), and fully discharged state). The bond lengths and BVS information are displayed in the table below based on the Rietveld refinement results.

For the bond length, the Ni-O bond decreases owing to the oxidation of Ni³⁺, and the Li-O bond is enlarged correspondingly. After discharge, all bond lengths can return to values similar to the pristine states. Regarding the BVS calculation, the predicted oxidation states of Ni at TM layers are 2.95, 2.86, and 2.87 for 108, 114, and 120, respectively. Based on the edge position of Ni in XANES, the order for the oxidation states of Ni is 120>108>114. The difference originates from the fact that the XANES results represent all Ni in the materials and the BVS predicts the oxidation states of Ni in TM layers. This can be further proved by the consistency between the two methods in the oxidation states of Mn: the oxidation states of Mn are predicted to be 3.54, 3.42, and 3.44 for 108, 114, and 120, respectively. They are in line with the Mn K-edge XANES results because Mn is believed to be only situated in the TM layers. Therefore, the BVS values for Ni in the TM layers provide further structural information for pristine states. Most of the Ni ions in TM layers of 108 are Ni³⁺ and the numbers of Ni³⁺ ions are decreased for 114 and 120. The above discussions on bond lengths and BVS information have been added in Supplementary Note 5.

The polyhedral distortion is not considered here because of the difficulties in obtaining the detailed bond lengths for each pair of Ni-O bonds. Normally there is a cooperative Jahn-Teller distortion for LiNiO₂ when Li ions are partly removed from the lattice, accompanied by the formation of the monoclinic phase. However, such a monoclinic distortion disappears when dopants are introduced, so the polyhedral distortion is believed to be very weak and will not be studied in this research.

	A	B	C	D	E	F	G	H	I	J	K
1		Ni-O bond	Li-O bond	Li@Li	Ni@Li	Li@TM	Ni@TM	Mn@TM	Nb@TM	Mo@TM	W@TM
2	108-pristine	1.9482(15)	2.1525(19)	0.938(2)	1.560(3)	1.630(3)	2.954(5)	3.540(6)	5.426(9)	5.368(9)	5.515(9)
3	108-charge (81% SoC)	1.8758(21)	2.1833(28)	0.863(3)	1.435(4)	1.982(5)	3.592(8)	4.305(10)	6.598(15)	6.527(15)	6.706(15)
4	108-discharge	1.9341(15)	2.1724(18)	0.889(2)	1.478(3)	1.693(3)	3.069(5)	3.678(6)	5.637(9)	5.577(9)	5.730(9)
5											
6	114-pristine	1.9608(15)	2.1339(18)	0.987(2)	1.640(3)	1.575(3)	2.855(5)	3.422(6)	5.244(9)	5.188(9)	5.330(9)
7	114-charge (81% SoC)	1.8874(21)	2.1637(28)	0.910(3)	1.513(5)	1.921(5)	3.481(8)	4.172(10)	6.395(10)	6.326(15)	6.499(15)
8	114-discharge	1.9504(15)	2.1474(18)	0.951(2)	1.581(3)	1.620(3)	2.936(5)	3.519(6)	5.394(9)	5.336(9)	5.482(9)
9											
10	120-pristine	1.9592(23)	2.1257(27)	1.009(3)	1.677(5)	1.582(4)	2.868(7)	3.437(9)	5.268(13)	5.211(13)	5.354(13)
11	120-charge (81% SoC)	1.8897(29)	2.1696(38)	0.896(4)	1.489(6)	1.909(6)	3.460(11)	4.147(13)	6.536(20)	6.288(20)	6.460(21)
12	120-discharge	1.9400(22)	2.1465(27)	0.954(3)	1.585(5)	1.666(4)	3.020(7)	3.620(9)	5.548(14)	5.488(14)	5.638(14)

Comment # 6: Fig.3 The structural model needs to be validated and discussed by other methods (EXAFS, PDF).

Our reply: Thanks for the suggestion. The structure generated by diffraction were fitted with the software EXAFS Neo. The fits were done in k-range from 2.25 to 15 (Ni K-edge). The paths with a distance of max. 3.3 Å and an importance of more than 10% were used for the fit. A primary fitting result for the 108 sample is shown below. You can see a good agreement between the original spectrum and the fitted one.

The fitting was down with the EXAFS-NEO:

<https://www.sciencedirect.com/science/article/pii/S0169433221001355?via%3Dihub>.

However, notice that there are lots of uncertainties and optimization during fitting.

Therefore, it is only an exhibition of the possibilities and a primary verification of the structural model.

Comment # 7: Fig.4 It is necessary to consider the characteristic improvement factor by the discharge cutoff potential.

Our reply: Thanks for the good suggestion. The improvement in cycling stability originates from the sluggish kinetics of the oxygen redox. As discussed by the newly added characterizations dQ/dV profiles (Fig. S16), *ex situ* XAS (Fig. S15), *in situ* XAS (Fig. S14), and the lattice changes on discharge (Fig. S11), oxygen and Ni redox both exist during Li insertion and removal processes. At the end of the discharge, oxygen reduction dominates the mechanism behind the Li insertion and the extra capacity. The extension of the discharge voltage range helps realize a more complete oxygen redox thus increasing the reversibility of the cathodes. The capacity retention of all cathodes has been evaluated under a voltage range of 2.0 V- 4.3 V (Fig. S7). HD-LNMO-120 has the largest oxygen redox activity so it benefits a lot from the prolonging of the lower cut-off

voltage. On the contrary, the cycling performance of RM has been decreased because extending the voltage range only induces extra side reactions. The corresponding discussion has been added from page 20 to page 22, which now becomes the main part of the 'Structural stability' part.

Comment # 8: Fig,4 a, b Numerical values for the vertical axis and caption need to be added.

Our reply: Thanks for the suggestion. Now the numerical values are added in Fig. 4a, b.

Comment # 9: Since the properties of Li composition 1.2 are good, it is necessary to investigate with a higher composition. It is also not clear why 1.2 is better.

Our reply: The excellent battery performance of HD-LNMO-120 is rationalized by an increased contribution of oxygen redox in the total redox mechanism. The oxygen redox helps decrease the lattice changes on charge, especially at the higher voltage regions (above 4.1 V, suggested by the *in situ* XRD results), because oxygen redox generally just causes O-O bond shortening rather than collective Ni-O bond length change. Above discussions on oxygen redox can be found in the new 'structural stability' part. Those suppressed changes in lattice parameters and lattice volume improve the stability of the cathodes. The abrupt volume changes and highly anisotropic contraction in the *c* direction can cause mechanical failure (cracking and pulverization) of the electrodes, thus an increased rate of capacity fading of the Ni-rich cathodes <https://pubs.acs.org/doi/10.1021/acs.chemmater.7b05269>. We synthesized a sample using 26% excess Li salts, namely HD-LNMO-126. The structure can be checked from the corresponding XRD pattern. The electrochemical performance was also evaluated. The specific capacity and the capacity retention are much worse than the samples investigated in the main text. The bad performance may be induced by the large extent of the impurity phase or the local defects, which is out of the research focus in this manuscript.

Comment # 10: Table S2 - S6 The unit of R factor needs to be added. %

Our reply: Thanks for the suggestion. '%' is added.